# Phenotypic plasticity in cell elongation among closely related bacterial species

Marie Delaby [1,8], Liu Yang[1,6,8], Maxime Jacq[1], Kelley A. Gallagher[1,7], David T. Kysela[1], Velocity Hughes[1,2], Francisco Pulido[3], Frederic J. Veyrier[3], Michael S. VanNieuwenhze [4] & Yves V. Brun [1,5] ✉

Cell elongation in bacteria has been studied over many decades, in part because its underlying mechanisms are targets of numerous antibiotics. While multiple elongation modes have been described, little is known about how these strategies vary across species and in response to evolutionary and environmental influences. Here, we use fluorescent D-amino acids to track the spatiotemporal dynamics of bacterial cell elongation, revealing unsuspected diversity of elongation modes among closely related species of the family *Caulobacteraceae*. We identify species-specific combinations of dispersed, midcell and polar elongation that can be either unidirectional or bidirectional. Using genetic, cell biology, and phylogenetic approaches, we demonstrate that evolution of unidirectional-midcell elongation is accompanied by changes in the localization of the peptidoglycan synthase PBP2. Our findings reveal high phenotypic plasticity in elongation mechanisms, with implications for our understanding of bacterial growth and evolution.

Cell elongation is a fundamental process in bacteria, underlying cell growth, morphogenesis, and division. Mechanisms of cell elongation are strictly regulated but respond dynamically to developmental and environmental cues[1–3]. For instance, spatiotemporal patterns of cell elongation are modulated over the cell cycle in many bacterial species. In others, nutrient starvation can trigger specialized modes of elongation, resulting in shape changes to improve nutrient uptake. Over the course of evolution, mechanisms of cell elongation have also diversified, as evidenced by the variety of cell shapes, life cycles, and mechanisms of growth seen in bacteria[4,5].

At the structural level, bacterial cell elongation is driven by the synthesis of the peptidoglycan (PG) cell wall, a meshwork of glycan strands crosslinked by peptide chains that confers structural integrity to the cell[3,6,7]. Synthesis of PG is essential to bacterial growth and division[8] and is catalyzed by the penicillin binding proteins (PBPs) and SEDS (Shape, Elongation, division, and sporulation) family proteins[9,10],

acting in concert with regulatory protein complexes called the elongasome (for cell elongation) and the divisome (for cell division). Although the PBPs, the elongasome and the divisome are widely conserved among bacteria, different species have been shown to employ distinct sets of proteins to assemble their elongasomes[11–16]. Such modularity and flexibility in the regulation of cell elongation at the molecular level may underlie the evolution and diversification of bacterial cell elongation mechanisms. Indeed, recent studies have found that disparate mechanisms of cell elongation can yield similar shapes: a rod-shaped cell can be generated by a dispersed, lateral elongation mode, or by zonal elongation from one or both cell poles or the midcell (Fig. 1a)[11,17–21]. Conversely, the same elongation mode can generate different cell shapes: spherical, ovoid, rod, and crescent shapes can be generated by bidirectional elongation from the midcell[22,23]. However, our understanding of the diversity of elongation modes in bacteria is primarily based on a few distantly related model

[1]Département de microbiologie, infectiologie et immunologie, Université de Montréal, Montréal, QC, Canada. [2]Synthesis by Velocity, Malmö, Sweden. [3]Bacterial Symbionts Evolution, Centre Armand-Frappier Santé Biotechnologie, Institut National de la Recherche Scientifique, Laval, QC, Canada. [4]Department of Chemistry, Indiana University, 800 East Kirkwood Avenue, Bloomington, IN, USA. [5]Department of Biology, Indiana University, 1001 E. 3rd St, Bloomington, IN, USA. [6]Present address: Biosphere Sciences and Engineering, Carnegie Institution of Washington, 813 Santa Barbara Street, Pasadena, California, USA. [7]Present address: Department of Microbiology, Cornell University, Ithaca, NY, USA. [8]These authors contributed equally: Marie Delaby, Liu Yang. ✉e-mail: yves.brun@umontreal.ca

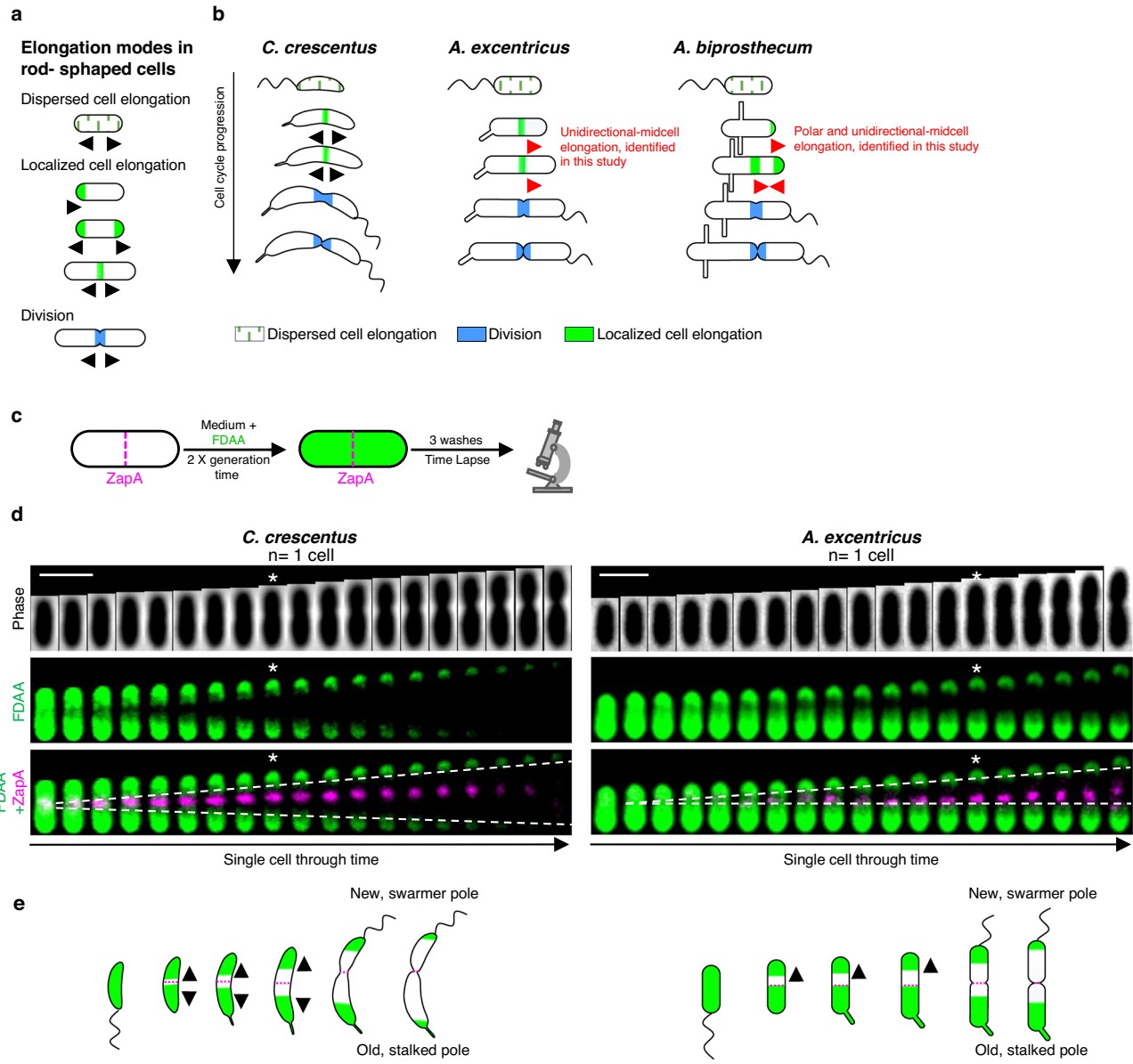

**Fig. 1 | Diversity of growth modes in rod-shaped bacteria and members of the *Caulobacteraceae* family. a** Schematic illustrating different growth modes observed among rod-shaped bacteria. Green lines represent lateral or dispersed elongation, solid green denotes different modes of localized cell elongation, and solid blue marks cell division. **b** Dimorphic cell cycles and growth modes in *C. crescentus*, *A. excentricus,* and *A. biprosthecum*. *C. crescentus* swarmer cells elongate by dispersed cell elongation (green lines) before bidirectional midcell elongation (green), followed by cell division (blue). *A. excentricus* swarmer cells also undergo dispersed elongation (green lines). However, this is followed by an uncharacterized, unidirectional midcell elongation towards the new pole (green) prior to bidirectional cell division (blue). *A. biprosthecum* cells elongate through yet another novel mode of elongation, a combination of polar and unidirectional midcell elongation. **c** Schematic depicting the pulse-chase experiment using FDAA. Whole-cell PG was labeled with FDAA (green) over two generations, followed by washes with PYE to remove free FDAA from the medium. Subsequent growth in the absence

of FDAA was observed using time-lapse microscopy. During the chase period, the loss of the FDAA signal corresponds to new PG synthesis/turnover. The magenta dashed line indicates the position of the ZapA fluorescent protein fusion, as a marker of the future division site. **d** Pulse-chase experiments using FDAAs in *C. crescentus* and *A. excentricus* cells carrying fluorescent fusions of ZapA. Images were taken every 5 min during the chase period. Kymographs show the loss of FDAA fluorescence as the cells grow. In *C. crescentus* (left), the FDAA BADA was used in combination with ZapA-mCherry. In *A. excentricus* (right), the FDAA TADA was used in combination with ZapA-sfGFP. Kymographs present the FDAA signal in green and the ZapA fluorescent fusion signal in magenta. White stars indicate cells starting division. See Supplementary Fig.1 for additional kymographs from each species, *n* = 3 biological replicates. Scale bars: 2 μm (cell length). **e** Schematic of FDAA signal loss in *C. crescentus* and *A. excentricus* cells. The magenta dashed line indicates the position of ZapA as a marker of the future division site. Green shading represents the old PG labeled with FDAA.

organisms, leading to the assumption that closely related species share the same elongation strategies. Consequently, the true diversity of bacterial cell elongation mechanisms may be greatly underappreciated, and our insights reveal little about the evolutionary mechanisms driving these differences.

To address this knowledge gap, our study explores how distinct elongation modes evolved within closely related, morphologically

diverse bacterial species from the *Caulobacteraceae* family[4,5,24]. Within this family, the well-studied model organism *Caulobacter crescentus* uses bidirectional elongation from the midcell[25] (Fig. 1b). By studying its related species, we discover two uncharacterized elongation modes – unidirectional midcell elongation in the species *Asticcacaulis excentricus*, and polar plus unidirectional midcell elongation *in Asticcacaulis biprosthecum*. Using a multidisciplinary approach that integrates live-

cell imaging with genetic and evolutionary analysis, we explore these elongation mechanisms. Our findings reveal that the evolution of these elongation strategies is associated with shifts in the spatial localization of a core elongasome enzyme. Specifically, we show that the penicillin-binding protein PBP2, a class B PBP, which localizes diffusely in *C. crescentus*, concentrates at the midcell in *A. excentricus*, where it serves as the transpeptidase enzyme driving unidirectional PG synthesis. This highlights how the regulation of conserved elongasome machinery can vary even between closely related species, potentially aligning with broader evolutionary pressures. Extending our analysis beyond the *Caulobacteraceae*, we found that *Rhodobacter capsulatus* shares the unidirectional midcell elongation pattern seen in *A. excentricus*, and we further infer from the phylogeny of the Alphaproteobacteria that bacterial cell elongation mechanisms display far greater phenotypic plasticity than previously anticipated. Ultimately, these results challenge the notion that closely related species share the same elongation mode. Instead, they reveal an unexplored diversity in elongation mechanisms, as cells respond to evolutionary forces to generate diversity in growth modes even at close evolutionary scales.

## Results

### *C. crescentus* and *A. excentricus* have different patterns of PG synthesis

The order *Caulobacterales* exhibits at least three elongation modes – dispersed, polar, and midcell[4], suggesting that it could be a good model for studying the evolution of cell elongation. The order comprises three families: the *Caulobacteraceae*, the *Hyphomonadaceae*, and the *Maricaulaceae*. The *Hyphomonadaceae* elongate polarly through budding[26–28], but the elongation modes in the *Maricaulaceae* and the *Caulobacteraceae* are unknown, with the exception of the model organism *C. crescentus*, which predominantly grows through bidirectional midcell elongation, with some dispersed elongation during early stages of the cell cycle (Fig. 1b).

To determine the elongation modes in other *Caulobacteraceae*, we analyzed the genus *Asticcacaulis*, comparing *C. crescentus* and *A. excentricus* using pulse-chase experiments with fluorescent D-amino acids (FDAAs). FDAAs are fluorescent dyes that are incorporated into the PG by PG transpeptidases, serving as highly effective tools to observe the dynamics of cell growth in various colors (in this study, we use the dyes BADA, HADA, and TADA, see "Methods")[29,30]. To track PG synthesis/turnover in real-time relative to the cell division site, we marked the division site in *C. crescentus* using ZapA-mCherry and in *A. excentricus* using ZapA-sfGFP[31], and performed pulse-chase experiments using FDAAs with complementary fluorophores. Briefly, we labeled whole-cell PG with FDAA (the "pulse") over two generations, washed the cells to remove free FDAA, and observed the cells during a period of growth in the absence of FDAA (the "chase") by time-lapse microscopy (Fig. 1c). During the chase period, the loss of FDAA labeling reveals spatial patterns of new PG incorporation[32].

In dimorphic bacteria such as *Asticcacaulis* and *Caulobacter* species, the new pole generated by division gives rise to a motile swarmer cell, while the old pole forms the larger, non-motile stalked cell, reflecting the species' characteristic asymmetric division. In *C. crescentus*, visualizing the retention and loss of the FDAA label throughout the cell cycle, we observed that the loss of FDAA labeling occurred from the division plane defined by ZapA-mCherry, and extended bidirectionally towards both cell poles (Fig. 1d **and** Supplementary Fig. 1a, b). Conversely, in *A. excentricus*, the loss of FDAA labeling originated from the division plane but moved predominantly towards the new pole of the dividing cell (Fig. 1d **and** Supplementary Fig. 1c, d). Later in the cell cycle, *A. excentricus* cells also began losing the FDAA signal on the old pole side of the ZapA-sfGFP signal once constriction was initiated (Fig. 1d,**white stars**). This loss of the FDAA signal on the old pole side of the division plane during constriction likely

corresponds to PG synthesis during septation and cell division. Together, these data indicate that PG synthesis/turnover for cell elongation is spatially localized close to the future site of cell division in both *C. crescentus* and *A. excentricus*. Furthermore, they indicate a variation in PG synthesis/turnover dynamics in these two closely related species, in which *C. crescentus* elongates bidirectionally and *A. excentricus* towards the new pole (Fig. 1b, e).

### *A. excentricus* elongates unidirectionally from the midcell towards the new pole

To further investigate the position and directionality of PG synthesis during midcell elongation in *A. excentricus*, we conducted sequential short-pulse FDAA labeling experiments with differently colored FDAAs. This approach allows us to track active sites of PG synthesis in growing cells based on the spatial pattern of incorporation of the sequentially applied dyes over time[32]. If the signal from the first FDAA (magenta) pulse appears on both sides of the second FDAA (green), PG incorporation can be inferred to occur bidirectionally, whereas if the first FDAA signal appears on one side of the second signal, it would indicate unidirectional elongation (Fig. 2a).

For these experiments, *A. excentricus* and *C. crescentus* cells were first grown in the presence of one FDAA (shown in magenta) for 5% of their generation time, washed to remove excess dye, and then subjected to a second pulse with a different FDAA (shown in green) (Fig. 2a). In *C. crescentus*, the first, magenta FDAA signal appeared on both sides of the second, green signal, which was approximately located at the midcell (Fig. 2b). To analyse these patterns quantitatively, we generated demographs, which are graphical summaries of fluorescence intensities, where cells are sorted by length along the y-axis, and aligned at the center by their midcell as a proxy for cell cycle progression. Demograph analysis of *C. crescentus* cells confirmed that their first FDAA signal was located on both sides of the second signal in all but the shortest (i.e., most newly divided) cells (Fig. 2b bottom-left and Supplementary Fig. 2b). For further quantification, the fluorescence intensities of the FDAAs were normalized and plotted against their relative positions along the cell length for a subset of cells, from roughly the middle of their cell cycles. As in the demograph, the magenta signal appeared on both sides of the green signal, confirming bidirectional midcell elongation in *C. crescentus* (Fig. 2b bottom-right). Notably, we also observed a slight asymmetry in fluorescence intensity, with a stronger signal on the old pole side. Given that *C. crescentus* divides asymmetrically into a swarmer and a stalked cell, and that the stalked cell is typically longer at the time of labeling, it may incorporate more PG during elongation. This could result in apparent differences in labeling while still maintaining bidirectional elongation.

Contrastingly, in *A. excentricus*, the first, magenta FDAA signal appeared only on one side of the midcell region, whereas the second, green signal appeared at the midcell region, demonstrating unidirectional PG synthesis (Fig. 2c and Supplementary Fig. 2b). To determine whether the unidirectional midcell elongation observed in *A. excentricus* consistently proceeds towards the new pole, we conducted dual short-pulse FDAA labeling experiments in *A. excentricus* cells in which the old poles were marked. For this, we utilized the dimorphic cell cycle of this species, which features regulated changes in morphology and surface adhesion within the context of the cell cycle[33,34]. Specifically, cells produce a holdfast polysaccharide adhesin at the old pole during cell cycle progression, which can be used as an old-pole marker through labeling with fluorescent wheat germ agglutinin (WGA) lectin. Dual short-pulse FDAA labeling in *A. excentricus* cells with WGA-labeled holdfasts showed that the first FDAA signal was always on the new-pole side of the second FDAA signal (Fig. 2d). These results demonstrate that *C. crescentus* has a bidirectional midcell elongation mode while *A. excentricus* has a unidirectional midcell elongation mode in which PG synthesis proceeds towards the new pole (Fig. 2e).

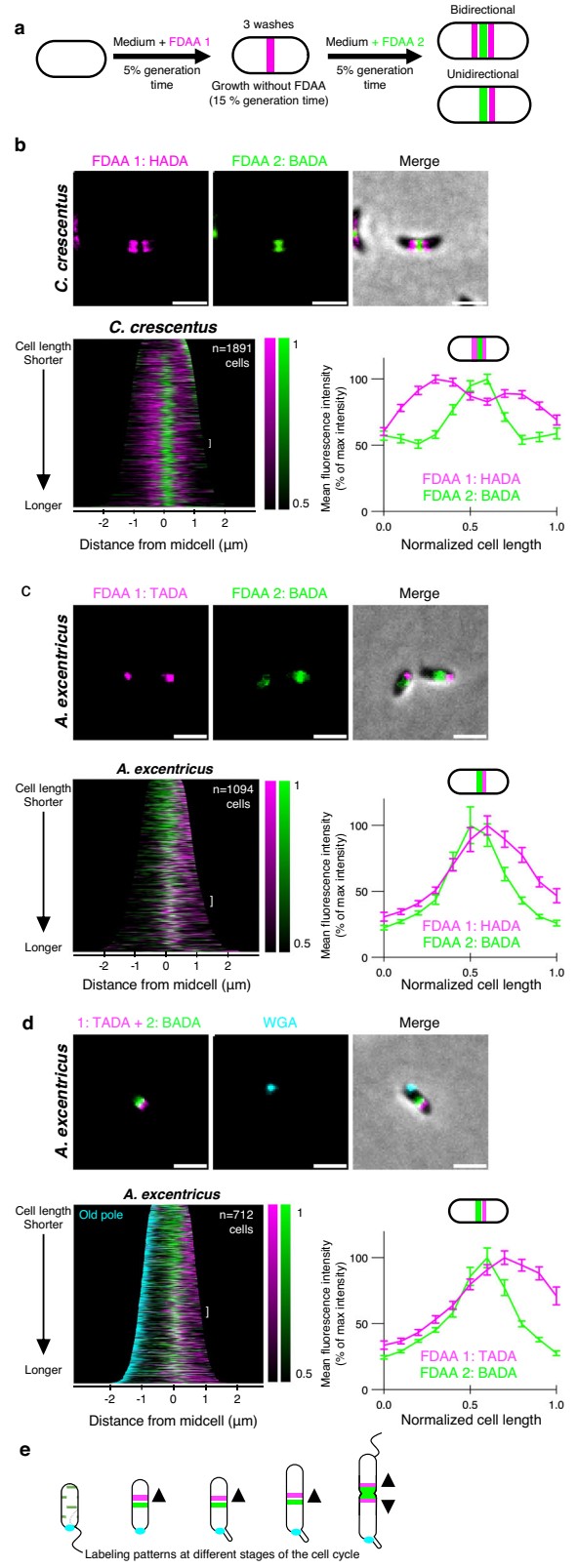

**Fig. 2 | Sequential FDAA labeling reveals that *C. crescentus* grows bidirectionally from the midcell, while *A. excentricus* grows unidirectionally towards the new pole. a** Schematic depicting the dual short-pulse experiment: sequential, dual short FDAA pulses show sites of active PG synthesis. Cells were first labeled with one FDAA (HADA in *C. crescentus* or TADA in *A. excentricus*, magenta) for 5% of their generation time, washed with PYE to remove the free FDAA, allowed to grow for 15 % of their generation time, and then labeled with a second FDAA (green) for 5% of their generation time, washed again, and imaged with microscopy. **b**–**d** Sequential FDAA labeling in *C. crescentus* (**b**) and *A. excentricus* **c**,**d**. *Top*: Representative images are shown (FDAA 1, FDAA 2, and merge, *n* = 3 biological replicates). Scale bars: 2 μm. *Bottom*: population-level demographs showing the localization of the fluorescence intensities of FDAA 1 and FDAA 2, and graphs showing the fluorescence intensity of the FDAAs against their relative positions along the cell length for 50 cells. In (**d**), the old pole in *A. excentricus* cells is additionally labeled with fluorescent WGA (cyan). In all demographs, cells are arranged by length, with 50% of maximum fluorescence intensities shown. Demographs in Panels (**b**,**c**) are oriented with the maximum fluorescence intensity of the second FDAA to the right, and in Panel **d** with the WGA-labeled old pole (cyan) to the left. In (**b**,**c**, and **d**), the white brackets show the 50 cells selected to plot the fluorescence intensities of the two FDAAs. The lines represent the mean values, with error bars showing the standard error of the mean (SEM). See Supplementary Fig. 2 for additional demographs. Source data are provided as a Source Data file. **e** Schematic illustrating PG synthesis during *A. excentricus* cell cycle. Swarmer cells undergo dispersed cell elongation (green dots). Stalked cells elongate unidirectionally from the midcell towards the new pole, with the first FDAA signal (magenta) located on the new pole side of the second FDAA signal (green). Predivisional cells exhibit bidirectional growth at the midcell, with magenta signals on both sides of the second FDAA signal (green). The old pole is indicated by the holdfast (cyan).

limited repertoire of PG synthases in *A. excentricus*: three class A PBPs, including two homologs of PBP1a (Astex_2378 and Astex_2994) and one homolog of PBP1c (Astex_0196); and two monofunctional class B PBPs – PBP2 and PBP3 (FtsI) (Supplementary Fig. 3a). This limited repertoire of PBPs in *A. excentricus*, and the conserved neighborhood architecture of the cell elongation loci in *C. crescentus* and *A. excentricus* (Supplementary Fig. 3b) suggested that the same canonical elongasome proteins may have evolved to generate different modes of elongation in the two species. Therefore, we hypothesized that localization of PBP2, an essential component of the elongasome, may play a role in the evolution of these different elongation modes.

To investigate the role of PBP2 in midcell elongation, we utilized fluorescent protein fusions to track its subcellular localization. We fused PBP2 to mCherry at its native locus in *A. excentricus* and performed fluorescence microscopy (See Supplementary Note 1 and Supplementary Fig. 8 for validation of the mCherry-PBP2 fusion). Across the population, PBP2 exhibited a patchy localization with enrichment near the midcell (Fig. 3a). Quantifying its localization in a large number of cells in a demograph, or using a population-wide heatmap of subcellular PBP2 localization using the holdfast as a polar marker (Fig. 3a), we found that PBP2 accumulates close to the midcell, offset towards the new pole, which is similar to the location of new PG synthesis. To further analyze whether PG synthesis at the midcell coincides with PBP2 localization, we labeled *A. excentricus* cells expressing mCherry-PBP2 with a short pulse of FDAA. Fluorescence microscopy and quantification using a population-wide heatmap and density map of the maximal fluorescence intensities of the mCherry-PBP2 and FDAA signals showed that the FDAA signal overlapped with the mCherry-PBP2 signal (Fig. 3b and Supplementary Fig. 4a, b). Together, these data indicate that PBP2 localization correlates with sites of PG synthesis during unidirectional midcell elongation in *A. excentricus*.

To further probe the link of PBP2's distinct localization in *A. excentricus* to its alternative mode of cell elongation, we investigated PBP2 localization in *C. crescentus*, which exhibits bidirectional

## The *A. excentricus* class B PG synthase PBP2 localizes to sites of new PG synthesis

To understand the evolutionary origins of unidirectional midcell elongation in *A. excentricus*, we investigated the potential molecular determinants driving this distinct elongation mode. Bioinformatic analysis using *C. crescentus* PBP2 and PBP1a as queries revealed a

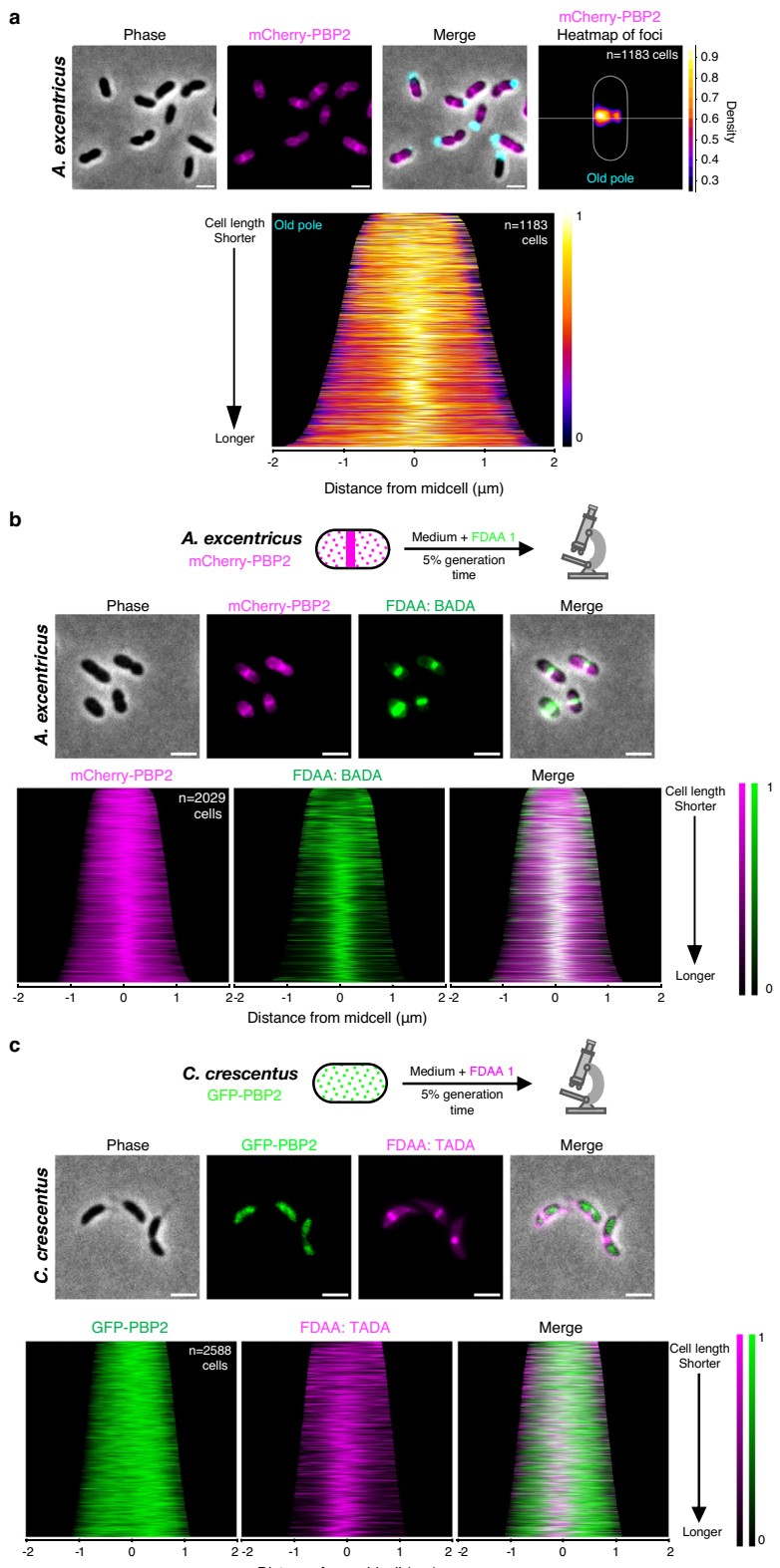

elongation from the midcell[25]. Using a GFP fusion to PBP2 at its native locus in *C. crescentus*, we observed that GFP-PBP2 displayed a dispersed distribution throughout the cell cycle in *C. crescentus*, consistent with previous studies[35], and distinct from PBP2 localization in *A. excentricus* (Fig. 3c and Supplementary Fig. 4c, d). To determine whether PBP2 colocalizes with sites of PG synthesis in *C. crescentus*, we performed short-pulse FDAA labeling in cells expressing GFP-PBP2. Fluorescence microscopy and population-level quantification of cells in a demograph revealed that, unlike in *A. excentricus*, *C. crescentus* PBP2 did not show significant enrichment at the midcell, and the FDAA labeling did not overlap with the GFP-PBP2 signal (Fig. 3c). This difference in the localization patterns of PBP2 in *C. crescentus* and *A.*

**Fig. 3 | Distinct PBP2 localization in *A. excentricus* and *C. crescentus* and its correlation with active PG synthesis. a** Subcellular localization of mCherry-PBP2 in *A. excentricus*. *Top:* Representative images are shown (*n* = 3 biological replicates). Left to right: Phase, mCherry-PBP2 fluorescence, and merged images. Scale bar: 2 μm. A heatmap of mCherry-PBP2 foci at the population level is displayed. In the heatmap, cells were oriented using the old pole labeled with WGA (cyan), with the white line indicating the midcell. *Bottom*: A demograph showing the localization of mCherry-PBP2 fluorescence at the population level, with cells arranged by length and oriented with the old pole towards the right. **b** Short-pulse FDAA (BADA) labeling of the *A. excentricus* mCherry-PBP2 strain. *Top:* A schematic depicting the experiment. Cells were labeled with 250 μM BADA for 5% of their generation time, fixed with 70% (v/v) ethanol, and imaged. *Middle:* Representative phase, fluorescence (mCherry-PBP2 and BADA), and merged images are shown (*n* = 3 biological

replicates). Scale bar: 2 μm. *Bottom:* Population-level demographs showing the localization of the fluorescence intensities of mCherry-PBP2, BADA, and their overlays. See Supplementary Fig. 4a for heatmaps and density maps of mCherry-PBP2 and BADA at the population level. **c** Short-pulse FDAA (TADA) labeling of the *C. crescentus* GFP-PBP2 strain. *Top:* A schematic depicting the experiment. Cells were labeled with 250 μM TADA for 5% of their generation time, fixed with 70% (v/v) ethanol, and imaged. *Middle:* Representative phase, fluorescence (GFP-PBP2 and TADA), and merged images are shown (*n* = 3 biological replicates). Scale bar: 2 μm. *Bottom:* Population-level demographs showing the localization of the fluorescence intensities of GFP-PBP2, TADA, and their overlays. See Supplementary Fig. 4b, c for additional fluorescence images and population-level demographs showing the differences in localization of PBP2 in *A. excentricus* vs. *C. crescentus*.

*excentricus* suggests that the relocalization of PBP2 may be a critical evolutionary step in the divergence of the elongation modes between these two species.

## PBP2 activity is required for unidirectional elongation at the midcell in *A. excentricus*

To understand the functional role of the essential protein PBP2 in the unidirectional midcell elongation of *A. excentricus*, we utilized the β-lactam antibiotic mecillinam, a specific inhibitor of PBP2 transpeptidase activity in *Escherichia coli*[36]. In *C. crescentus*, mecillinam causes cell bulging[37,38], likely through its inhibition of PBP2 activity (Supplementary Fig. 5a). Similarly, mecillinam treatment led to cell bulging in *A. excentricus*, suggesting that it interferes with cell wall synthesis in this species as well (Supplementary Fig. 5a). To determine the target of mecillinam in *A. excentricus*, we conducted competition assays with the β-lactam Bocillin FL, a fluorescent penicillin that covalently binds all PBPs. Using Bocillin gel analysis, we found that mecillinam predominantly inhibited binding of Bocillin to PBP2, even at high concentrations (100 μg ml$^{-1}$), while other PBPs in *A. excentricus* were only partially affected (Supplementary Fig. 5b). These results indicate that mecillinam is a suitable tool to investigate the role of PBP2 in cell elongation in *A. excentricus*.

To analyze the effect of PBP2 inhibition on cell elongation, we conducted an FDAA pulse-chase experiment in *A. excentricus* cells expressing ZapA-sfGFP, with or without mecillinam. We labeled whole-cell PG with FDAA over two generations without mecillinam, followed by a wash to remove excess FDAA, and a chase period with or without mecillinam for 120 min. In untreated cells, we found robust FDAA signal loss on the new pole side of the ZapA-sfGFP signal, consistent with unidirectional elongation at the midcell (Fig. 4a, left). In contrast, FDAA signal loss in mecillinam-treated cells was observed in bulges on the new pole side of the cell (Fig. 4a, right). To quantify the FDAA signal loss and cell bulging upon mecillinam treatment, we measured the subcellular localization of these bulges in relation to the fluorescence signals of ZapA and FDAA in a population-wide analysis. We plotted these features relative to the cell center, generating ShapePlots[39] for the whole population (Fig. 4b, right panels), as well as for four categories binned by cell length. Bulging was detected consistently on the new pole side of the midcell across all four categories, with its position shifting closer to midcell as cells progressed through the cell cycle and approached division (Fig. 4b, left panels). To analyze the patterns of FDAA labeling loss across populations of treated and untreated pulse-chased cells, we generated demographs and confirmed loss of signal towards the new pole of *A. excentricus* cells (Supplementary Fig. 5c, d). These analyses confirmed that bulging occurred primarily on the new pole side of the midcell in *A. excentricus*, i.e., at the site of midcell elongation (Fig. 4b, top panels). Furthermore, loss of the FDAA signal peaked at this site, suggesting that the bulging may be a result of abnormal PG synthesis at the midcell when PBP2 is inhibited (Fig. 4b, bottom panels). Together, these results suggest that PBP2 activity is required for the regulation of

elongasome activity at the midcell – both to promote unidirectional PG synthesis towards the new pole, and to prevent bulging.

To further analyze PBP2's role in the directionality of midcell elongation in *A. excentricus*, we conducted dual short-pulse experiments using two differently colored FDAAs in the presence or absence of mecillinam (Fig. 4c). In untreated cells, sequential FDAA labeling showed the first, magenta signal on only one side of the second, green signal, consistent with midcell elongation towards the new pole (Fig. 4c, left panels). However, in mecillinam-treated cells, both FDAA signals were distributed diffusely at the site of bulging, confirming the loss of directionality in PG synthesis upon PBP2 inhibition (Fig. 4c, right panels). Quantification of mean fluorescence intensities showed reduced FDAA incorporation in cells treated with mecillinam (Fig. 4d), consistent with the main transpeptidase of the cell being inhibited. Overall, these observations demonstrate that the inhibition of PBP2 activity severely disrupts the coordination of PG synthesis, leading to bulging at the site of the elongasome and loss of the unidirectional midcell elongation mode of *A. excentricus*. These findings confirm our hypothesis that PBP2's activity is required for unidirectional midcell elongation in *A. excentricus* and suggest that the enzyme may play a key role in organizing the directionality of elongasome activity in this species.

## Diverse modes of midcell elongation within and beyond the *Caulobacteraceae* family

To determine the elongation modes in other *Caulobacteraceae*, we analyzed other species in the genera *Asticcacaulis*, *Phenylobacterium*, *Brevundimonas*, and *Caulobacter* using pulse-chase experiments with FDAA (Fig. 5a). To analyze the patterns of FDAA labeling loss across populations of pulse-chased cells, we generated demographs for each species (Fig. 5b). Smaller cells of all six species showed a loss of signal at one of their poles, which likely corresponds to a recent division (Fig. 5b, white stars). In longer cells, the loss of FDAA labeling during the chase period was located near the midcell, corresponding to midcell elongation at the future site of cell division, or indeed to cell division itself (Fig. 5b). In *C. crescentus*, *C. henricii*, and *B. diminuta,* the loss of FDAA labeling was located near the midcell, progressing symmetrically towards both cell poles. This observation is consistent with PG synthesis in *C. crescentus* occurring bidirectionally from the midcell region in stalked cells[25] (Fig. 1b–d, Fig. 2, and Fig. 5c, d). In contrast, the loss of FDAA labeling during the chase period in *A. excentricus*, *A. biprosthecum,* and *P. conjunctum* started in the midcell region but trended towards the left of the demograph (Fig. 5b, red arrows).

The loss of FDAA predominantly towards one pole in *A. biprosthecum* and *P. conjunctum* suggests a unidirectional elongation mode in these species, similar to *A. excentricus*. To further investigate this, we performed pulse-chase FDAA labeling and time-lapse microscopy in *A. biprosthecum* (Fig. 6a). Interestingly, kymograph analysis in this species revealed not only unicellular midcell elongation towards the new

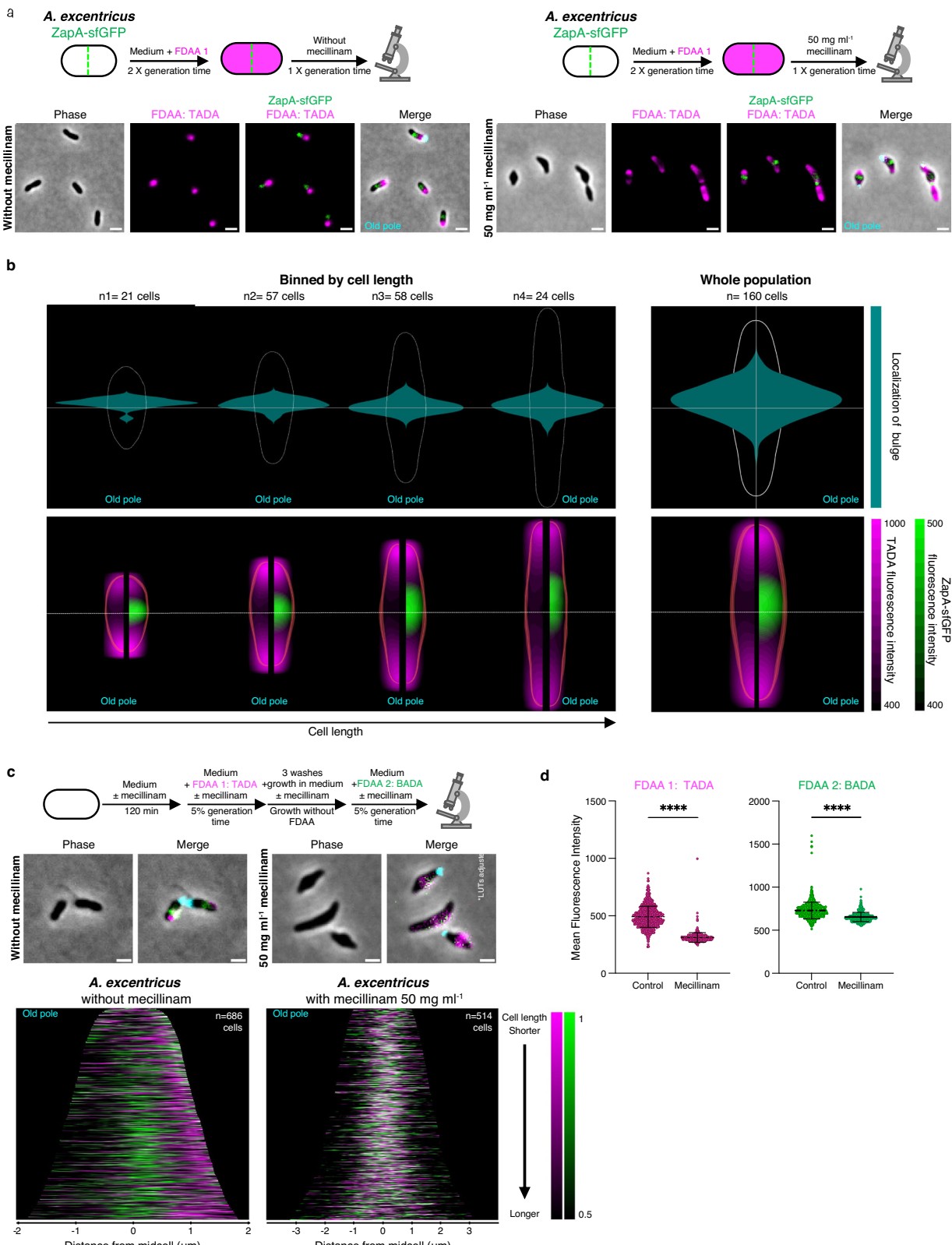

pole, as in *A. excentricus*, but also an additional site of elongation at the new pole itself (Fig. 6b and Supplementary Fig. 6). This observation suggests that *A. biprosthecum* elongates through yet another uncharacterized mode – a combination of polar elongation and unidirectional midcell elongation. To further analyze this mode of cell elongation, we sequentially added differently colored FDAAs during short pulses (5% of a generation time, Fig. 6c). Demograph and fluorescence profile analyses revealed that the second, green FDAA signal was present at both the pole and the midcell, while the first, magenta signal was located mainly at the pole. At the pole, the second green signal appeared at the tip, while the magenta signal was positioned just adjacent away from the tip, indicating apical elongation (Fig. 6d, e and Supplementary Fig. 6). Interestingly, the population-level demographs showed that the FDAA signals at the pole were present throughout the

**Fig. 4 | PBP2 is responsible for unidirectional PG synthesis at the midcell in *A. excentricus*. a** Schematic depicting pulse-chase experiments in *A. excentricus* ZapA-sfGFP cells with (*right*) or without (*left*) mecillinam treatment. Whole-cell PG was labeled with TADA (magenta) over two generations, followed by washing and growth with or without mecillinam (50 µg ml⁻¹) over one generation before imaging. Representative phase, fluorescence (TADA and TADA overlaid with ZapA-sfGFP), and merged images with WGA fluorescence are shown (*n* = 3 biological replicates). See Supplementary Fig. 5c for population-level demographs. Scale bar: 2 µm. **b** ShapePlots of *A. excentricus* cells after an FDAA pulse-chase experiment, with mecillinam. *Top:* Shape plots showing bulge localization. *Bottom:* ShapePlots show ZapA-sfGFP and FDAA signal loss. Each ShapePlot is divided longitudinally (black line) with FDAA signal loss on the left and overlaid with ZapA-sfGFP on the right. ShapePlots show four categories of cells binned by cell length (*left*) and the entire population of 160 cells (*right*) oriented using WGA-labeled holdfast (old pole at the bottom). Horizontal white lines represent the midcell. **c** *Top:* Schematic of a dual

short-pulse experiment with or without mecillinam (50 µg ml⁻¹). *A. excentricus* cells were grown over 120 min with or without mecillinam, then labeled with TADA (magenta), washed, allowed to grow, and labeled with BADA (green). Cells were washed again and imaged with microscopy. *Middle:* Representative phase and merged images from both treatment conditions (*n* = 3 biological replicates). Merged images show phase contrast overlaid with fluorescence signals from the two FDAAs and WGA-labeled holdfast (cyan). Scale bar: 2 µm. Look-up tables (LUTs) were adjusted for each condition to have a visible FDAA signal. *Bottom:* Population-level demographs showing the fluorescence intensities of both FDAA signals, with and without mecillinam. Cells were arranged by length, with the old pole to the left. 50% of the maximum fluorescence intensities are shown. **d** Dot plots graphs showing quantification of fluorescence intensities for each FDAA pulse, with or without mecillinam (*n* = 686 cells control and *n* = 514 mecillinam treated-cells). **** *P* < 0.0001, unpaired two-tailed t-test with Welch's correction. Error bars show the SEM. Source data are provided as a Source Data file.

cell cycle, whereas signals at the midcell appeared only in longer cells. This pattern indicates a sequence of polar cell elongation early in the cell cycle, followed by unidirectional midcell elongation closer to the time of division. The observation of both polar and unidirectional midcell elongation modes in *A. biprosthecum* underscores the diverse range of cell elongation mechanisms within the *Caulobacteraceae* family.

Given the diversity of elongation modes observed even within closely related species (Fig. 5b–d), we questioned whether unidirectional midcell elongation was a rare occurrence limited to the *Caulobacteraceae* family or if it could be observed in other species as well. We therefore extended our investigation to the more distantly related *R. capsulatus*, a member of the alphaproteobacterial order *Rhodobacterales*. We performed pulse-chase FDAA labeling and time-lapse microscopy as before (Fig. 6a, f and Supplementary Fig. 7). Kymograph analysis showed that the loss of FDAA labeling in *R. capsulatus* originated from the division plane and predominantly moved unidirectionally towards one of the cell poles, suggesting unidirectional midcell elongation. To further investigate the *R. capsulatus* cell elongation mode, we performed a dual short-pulse experiment and with two differently colored FDAAs (Fig. 6c). Demograph and fluorescence profile analyses revealed that the first FDAA signal in *R. capsulatus* was only on one side of the second FDAA signal, towards one of the cell poles (Fig. 6g, h). These results show that *R. capsulatus* also exhibits unidirectional elongation from the midcell, similar to the pattern observed in *A. excentricus*. This finding indicates that unidirectional midcell elongation is not restricted to the *Caulobacteraceae* family and suggests that this growth mode may be more widespread within the Alphaproteobacteria.

Overall, the diversity of cell elongation modes identified in this study suggests that there may be more undescribed modes to discover among bacteria. Our findings also emphasize the need for further exploration of the evolutionary transitions, such as PBP2 relocalization, that may underlie these differences.

## Discussion

The existence of different elongation modes in bacteria has been known for decades[40], but the mechanisms behind their evolution have remained unknown, largely due to research bias towards a few, distantly related model organisms. In parallel, there has been an assumption that closely related species share similar elongation mechanisms, hampering the study of divergence at shorter evolutionary scales. In this study, we challenge this assumption by revealing significant differences in the elongation strategies of closely related species within the *Caulobacteraceae* family. Our findings indicate that evolutionary changes in cell elongation can occur more frequently and at shorter evolutionary timescales than previously thought, opening new avenues for research into the evolution of bacterial growth and morphogenesis.

Among the *Caulobacteraceae*, the diversity and novelty of elongation modes we observe - ranging from bidirectional midcell elongation in *C. crescentus, C. henricii, and B. diminuta*, to the unidirectional midcell elongation in *A. excentricus* and *P. conjunctum*, as well as the combination of polar and unidirectional midcell elongation in *A. biprosthecum* (Figs. 1b, 5, 6), indicates remarkable evolutionary flexibility. They suggest a capacity for closely related species to adapt their elongation mechanisms, possibly in response to evolutionary pressures such as nutrient availability, morphological constraints, competition, predation, etc. It is already known that bacteria can regulate their elongation patterns in response to environmental changes. For instance, *Streptomyces* cells can shift their elongation mechanisms in response to metabolic cues, allowing exploratory growth[41,42]. Meanwhile, *Salmonella typhimurium* encodes two differentially regulated elongasomes, one of which is specialized for pathogenesis, functioning under acidic, intracellular conditions, with slower PG synthesis that may help coordinate cell elongation with host metabolism (although the spatial pattern of elongation remains the same)[43]. Many other bacterial species suppress the highly conserved divisome, using only elongation to produce a filamentous morphology to overcome a range of environmental constraints[44–46]. Finally, within the *Caulobacteraceae*, stalk elongation is a well-known response to nutrient limitation, although this specialized mode of PG synthesis at the stalk site is independent of the core elongasome that drives vegetative growth in the cell[33,47,48]. It would be intriguing to explore whether *A. excentricus* and related species similarly adjust their core elongation modes in response to environmental conditions. Future studies on this topic may shed light on the selective pressures that have led to the divergent elongation modes among the *Caulobacteraceae*.

The plasticity of cell elongation strategies is evident not only under different environmental conditions but even within a single cell under different regulatory states. For instance, among the *Caulobacteraceae*, the transition from dispersed elongation in the swarmer cell to localized cell elongation at the midcell (Fig. 1) demonstrates how elongasome components may be modulated by the cell cycle. Even in *E. coli*, which predominantly grows through dispersed elongation along its lateral walls, the elongasome-specific PG synthase PBP2 shows a cell-cycle dependent enrichment at the midcell prior to division, coinciding with a burst of midcell elongation[49]. Gram-positive coccoid bacteria such as *S. aureus* offer yet another example of the flexibility of the PG synthesis machinery, which functions both peripherally and for septal sidewall synthesis in these bacteria[50]. These observations underscore that elongation modes are not static in cells, but rather dynamically adaptable through regulatory networks controlling cell growth. Evolutionary forces may drive changes in bacterial elongation by modulating existing molecular mechanisms rather than by introducing new ones.

In this study, we provide evidence of previously uncharacterized cell elongation modes beyond the *Caulobacteraceae* (Fig. 5c). Indeed,

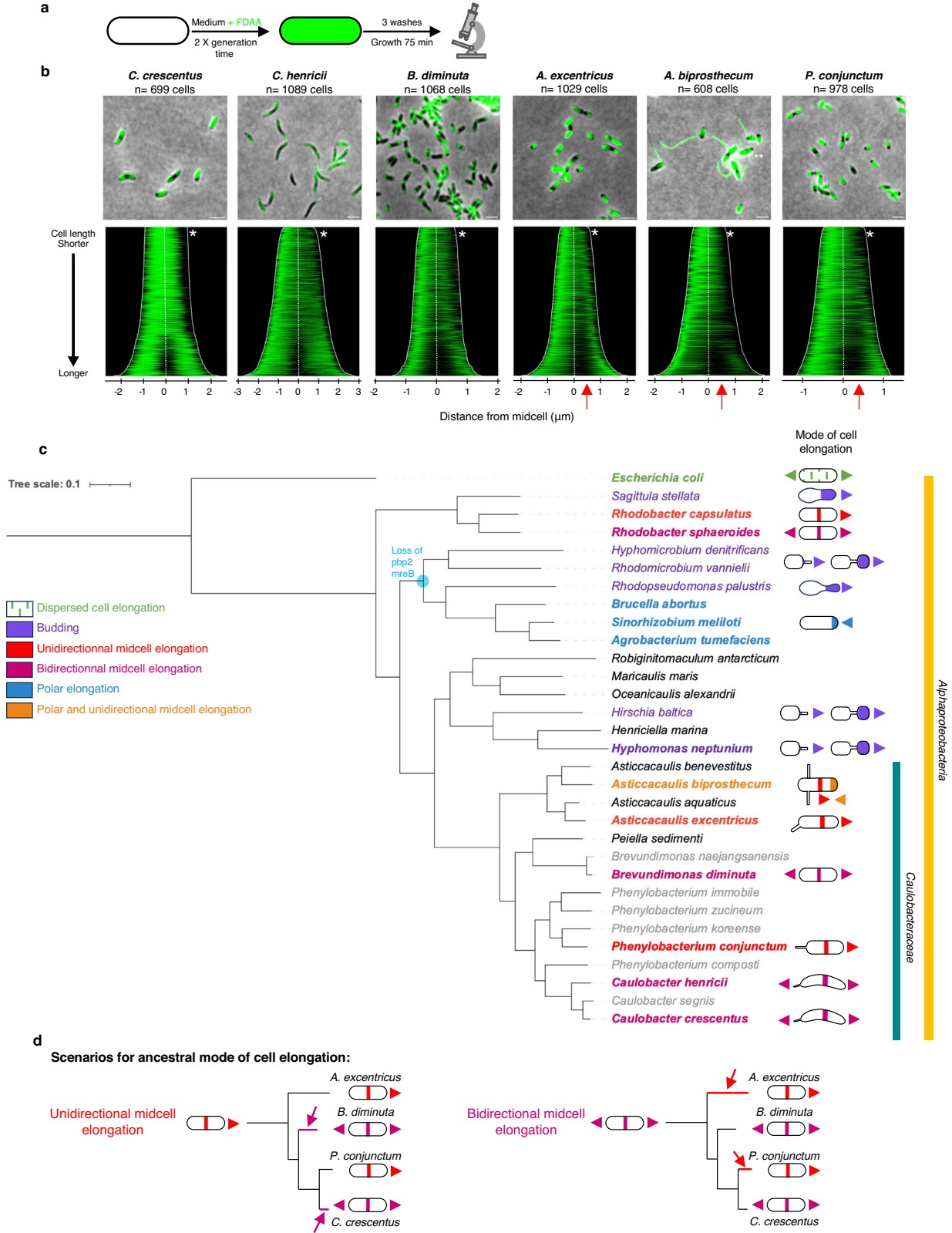

we discovered that *R. capsulatus* demonstrates unidirectional midcell elongation similar to *A. excentricus* (Fig. 6f–h). To understand the evolutionary context of this surprising discovery, we searched the literature for previous reports of elongation modes among the *Rhodobacterales* and found that *Rhodobacter sphaeroides* demonstrates bidirectional midcell elongation[51]. Meanwhile, in another branch of the Alphaproteobacteria, the loss of the key elongation proteins MreB and

PBP2 among members of the *Rhizobiales* has coincided with the evolution of polar elongation (Fig. 5c), further emphasizing the evolutionary plasticity of bacterial elongation systems[11]. Nonetheless, the conservation of elongasome clusters such as *pbp2/rodA* across species with different elongation modes, such as *E. coli, C. crescentus*, and *A. excentricus* (Supplementary Fig. 3), suggests that evolutionary changes in elongation modes stem not only from changes in gene content but

**Fig. 5 | Diversity of cell elongation modes in members of the *Caulobacteraceae* family. a** Schematic depicting the pulse-chase experiment using FDAAs. Whole-cell PG was labeled with TADA for two generations, washed to remove excess FDAA, followed by a chase period of 75 min before imaging. **b** Representative merge images are shown (*n* = 3 biological replicates). Demograph analysis of pulse-chase experiments in WT cells of *C. crescentus*, *C. henricii*, *P. conjunctum*, *B. diminuta*, *A. excentricus*, and *A. biprosthecum*. Cells are arranged by length, with each cell oriented so that the pole with the maximum fluorescence intensity is to the left. White stars indicate signal loss at one pole. Red arrows highlight unidirectional midcell elongation in *A. excentricus*, *A. biprosthecum*, and *P. conjunctum*, as observed by greater FDAA signal loss towards the right of the demograph. Scale bar: 2 μm. **c** Phylogenetic tree of representative species from the Alphaproteobacteria, which includes the family *Caulobacteraceae*. Taxon label colors correspond to different modes of cell elongation: dispersed cell elongation (green), unidirectional midcell elongation (red), polar elongation (blue), budding (violet), polar and unidirectional midcell elongation (orange and red), bidirectional midcell elongation (magenta), binary fission (black), and unknown (gray). Species for which the cell elongation mode has been studied using FDAAs, TRSE, or other methods are highlighted in bold. The node where *pbp2* and *mreB* (and the associated *mreCD* and *rodA* genes) are predicted to have been lost within the *Rhizobiales* is highlighted in blue. The tree, based on a concatenation conserved protein-coding gene sequences, is fully supported, with posterior probabilities of 1 for all clades. See the Methods section for details on phylogenetic reconstruction and refer to Supplementary Table S4 for genome IDs. **d** This schematic, derived from a pruned version of the phylogenetic tree in Fig. 5a to highlight the species of interest, illustrates the identified modes of cell elongation and the possible number of transitions. The transitions are depicted assuming two scenarios: unidirectional cell elongation as the ancestral state (*left, red*) or bidirectional midcell elongation as the ancestral state (*right, magenta*). Colored lines and arrows (*red or magenta*) indicate where the transitions might have occurred.

also from differences in the regulation, localization, or activity of these conserved proteins.

Here, in characterizing the molecular determinants of unidirectional midcell elongation in *A. excentricus*, we discovered that the PG synthase PBP2 is central to the regulation of this elongation strategy. In contrast to its dispersed localization in *C. crescentus*, PBP2 in *A. excentricus* concentrates at the midcell and is required for unidirectional PG synthesis. Upon inhibition of PBP2, cells exhibit abnormal PG synthesis, leading to bulging at the midcell elongation site. These results suggest that changes in the specific localization of key enzymes such as PBP2 may be associated with the evolution of distinct cell elongation strategies, enabling bacteria to respond to broader pressures acting on morphology and growth. Notably, in *C. crescentus*, PBP2 shifts to the midcell under osmotic stress, illustrating that its localization is subject to regulation both by environmental cues and evolutionary processes[35].

Beyond PBP2, other proteins like FtsZ play versatile roles in regulating localized PG synthesis across bacteria[52]. In *C. crescentus*, FtsZ depletion results in dispersed cell elongation[25], while in *Bacillus subtilis*, FtsZ can position itself at the midcell during vegetative growth or closer to the poles for sporulation[53], highlighting its flexibility in regulating PG synthesis for different growth modes. Similarly, DivIVA exemplifies how conserved proteins can acquire different functions in different bacterial lineages. In Actinobacteria, DivIVA is a key determinant of polar growth[54–57], directing PG synthesis at the cell poles, whereas in *B. subtilis*, it primarily functions in division site selection rather than elongation[58,59]. Recent work in *S. pneumoniae* has further demonstrated that DivIVA plays a central role in coordinating septum splitting and peripheral PG synthesis, ensuring the maintenance of elongation throughout the cell cycle[60]. This underscores the evolutionary plasticity of bacterial growth regulators and raises the possibility that other elongation proteins, such as PBP2, may have undergone similar adaptations to distinct cellular and evolutionary contexts. Another interesting example is the conserved outer membrane lipoprotein PapS, which in *Rhodospirillum rubrum* forms molecular cages that confine elongasomes to induce asymmetric cell elongation[61]. The diversity of mechanisms by which bacteria spatially regulate growth using conserved proteins supports the idea that evolution likely acts by altering the function of existing proteins rather than by introducing new genes. While several mechanisms could contribute to PBP2 localization in *A. excentricus*, including regulation through interactions with divisome or elongasome components or specific features of the PG itself, its precise positioning remains to be determined. A recent study in *E. coli* suggests that PBP2 may play a pivotal role in cell elongation, potentially acting as a trigger for the process. This work revealed that *E. coli* PBP2 exists in three diffusion states: free diffusion, immobile, and processively moving. Notably, the proportion of stationary and translocating PBP2 molecules remained stable even when PBP2 levels increased, and their binding was unaffected by the inactivation of other elongasome components. This suggests that MreB filaments and other elongasome members are not the primary binding substrates for PBP2. Instead, these findings support a model in which PBP2 acts as a key driver of elongation, potentially responding to specific features of the PG matrix to recruit the rest of the machinery for PG synthesis initiation[62]. In the future, it will be of interest to determine what scaffolding mechanisms are at play in *A. excentricus* and *A. biprosthecum*, driving the unique localization and directionality of PBP2 in their elongasomes.

The observation of diverse elongation mechanisms among bacteria raises the fundamental question of the function of such diversity. Why do some species like *A. excentricus* evolve unidirectional elongation, while others maintain dispersed, bidirectional, or polar growth mechanisms? What evolutionary factors drive these changes? It is clear that midcell and polar elongation are widely distributed in the Alphaproteobacteria (Fig. 5c), suggesting that localized modes of elongation may be ancestral within this class. However, the distribution of elongation modes in our phylogenetic analysis of the *Caulobacteraceae* (Fig. 5c, d) suggests that their evolution is likely shaped by multiple, independent events, implying a high degree of phenotypic plasticity in the regulation of localized elongation modes. In particular, the midcell, directional elongation identified in the current study is observed in *Asticcacaulis* and *Phenylobacterium*, but absent in members of *Caulobacter* and *Brevundimonas*, indicating that this growth mode has been gained and/or lost at least twice among the *Caulobacteraceae* (Fig. 5d). In light of this observed phenotypic diversity, it is intriguing to consider whether localized elongation itself may be a broadly adaptive strategy that bacteria have evolved, allowing greater plasticity under changing evolutionary pressures than dispersed elongation (although these co-exist in many species). A deeper understanding of the diversity of elongation modes among other branches of bacteria should provide further insights into the prevalence and ancestry of localized, dispersed, and septal modes of elongation, helping to identify selective pressures that shape the evolution of growth mode strategies.

In addition, the evolution of distinct elongation modes has implications for the evolution of bacterial morphology. *C. crescentus*, *A. excentricus*, and *A. biprosthecum* display striking morphological differences, especially in the positioning of their stalks[34]. This raises the possibility that their distinct elongation modes and stalk positioning mechanisms may have co-evolved. In all three species, the site of stalk synthesis corresponds to the region of the oldest PG in the swarmer cell, as it differentiates into a stalked cell. Could the elongation mechanisms of these species thus be driving the placement of their cellular structures? If future studies find this to be true, it would suggest that elongation modes and cellular morphologies are intricately linked, with changes and selective pressures on one likely to influence the other over evolutionary time.

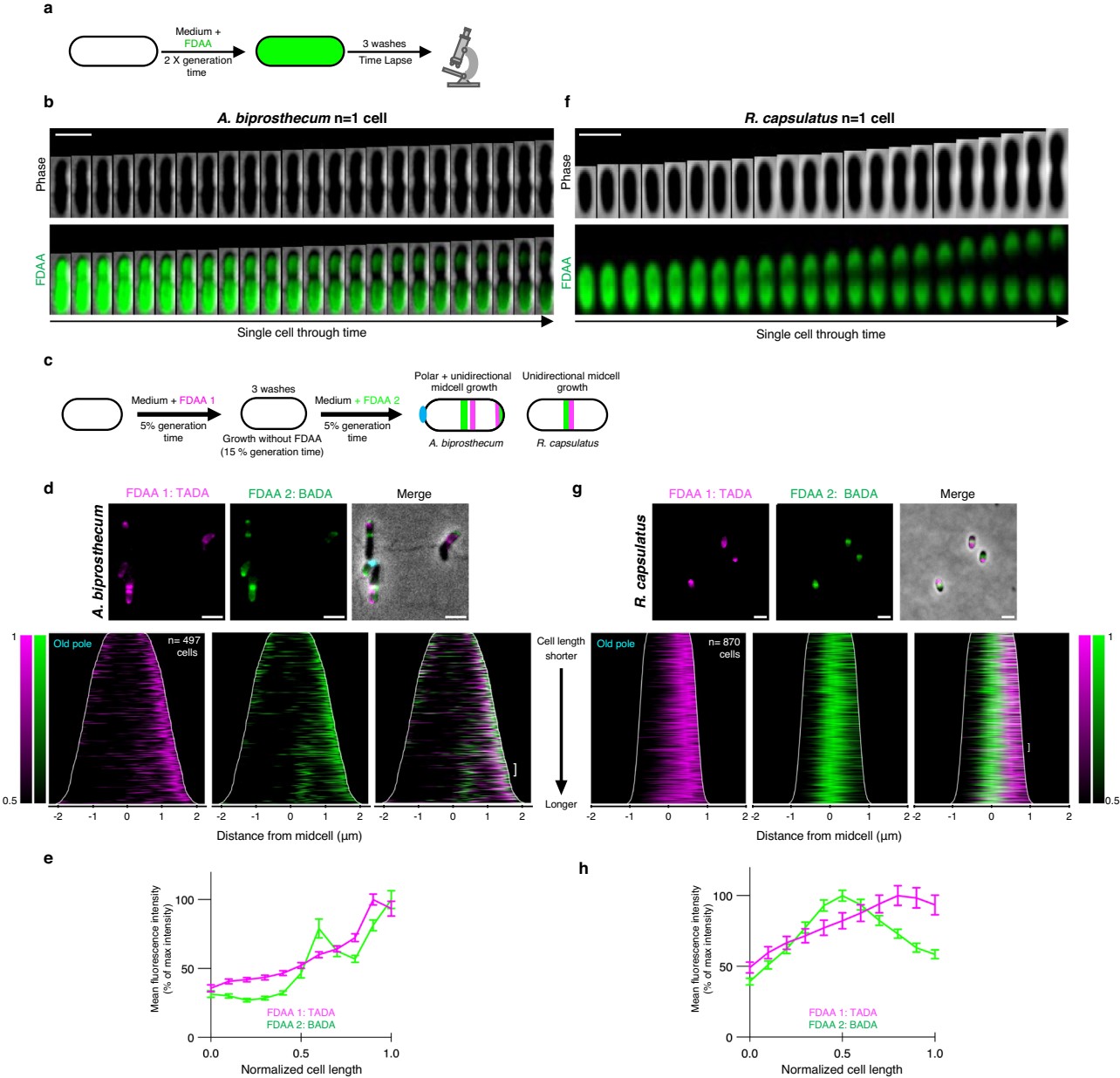

**Fig. 6 | FDAA labeling experiments demonstrate polar and unidirectional midcell elongation in *A. biprosthecum* and unidirectional midcell elongation in *R. capsulatus*. a** Schematic depicting the FDAA pulse-chase experiments in *A. biprosthecum* and *R. capsulatus*. Whole-cell PG was labeled with 500 μm TADA (green) over two generations, followed by washes to remove free FDAA from the medium. Subsequent growth in the absence of the FDAA was followed by time-lapse microscopy. **b,f** Kymographs of the pulse chase experiments showing the loss of FDAA fluorescence during the chase period in (**b**) *A. biprosthecum* and (**f**) *R. capsulatus*. Images were taken every 5 min (*n* = 3 biological replicates). Scale bars: 2 μm. See Supplementary Fig. 6 and 7 for additional kymographs. **c** Schematic depicting the dual short-pulse experiment. Cells were first labeled with TADA for 5% of their generation time, washed with fresh media, allowed to grow without FDAA, and then labeled with BADA for 5% of their generation time. Cells were then washed again and imaged with microscopy. **d,g** Representative images and demographs showing the fluorescence intensity of both FDAA signals in (**d**) *A. biprosthecum* and (**g**) *R. capsulatus* (*n* = 3 biological replicates). In the demographs, cells were arranged by length, with the old pole (labeled with WGA) to the left in *A. biprosthecum*, and with the maximum fluorescence intensity of the first FDAA to the right in *R. capsulatus*. 50% of the maximum fluorescence intensities are shown. The white brackets indicate the 50 cells selected to show the fluorescence profiles of the two FDAAs in (**e** and **f**). Scale bars: 2 μm. See Supplementary Figs. 6 and 7 for additional demographs. **e, h** Fluorescence intensity profiles of FDAA 1 (TADA, magenta line) and FDAA 2 (BADA, green line) in (**e**) *A. biprosthecum* cells and (**h**) *R. capsulatus* cells, plotted from *n* = 50 cells for both species. Points were selected along the medial axis of each cell, and the normalized signal was plotted relative to position along the cell length. The lines represent the mean values, with error bars showing the standard error of the mean (SEM). Source data are provided as a Source Data file.

In summary, the fundamental finding of this study is that bacterial elongation modes are more diverse than previously thought, even at relatively short evolutionary scales. This challenges the assumption that closely related species share similar elongation modes – an assumption that has prevailed largely because we have not had the right tools to explore the diversity of bacterial elongation modes, even at close evolutionary distances from well-studied model organisms. Combining spatiotemporal analyses of PG synthesis using FDAAs alongside genetic and evolutionary approaches now allows a reexamination of previous assumptions about cell elongation modes across bacterial clades. Consequently, we expect these results to be at the forefront of a paradigm shift in

our understanding of the diversity of bacterial cell elongation, its regulation, and evolution.

## Methods

### Bacterial strains and growth conditions

All bacterial strains used in this study are listed in Supplementary Table S1. *C. crescentus, B. diminuta,* and *C. henricii were grown at* 30 °C in Peptone Yeast Extract (PYE) medium[63]. *A. excentricus, A. biprosthecum,* and *P. conjunctum* were grown at 26 °C in PYE. *R. capsulatus* SB1003 cells were grown at 30 °C in PYS (3 g l$^{-1}$ peptone, 3 g l$^{-1}$yeast extract, 2 mM MgSO$_4$, 2 mM CaCl$_2$) medium[64]. The culture medium was supplemented with antibiotics as necessary at the following concentrations (µg ml$^{-1}$; liquid/solid medium): spectinomycin (50/100), kanamycin (5/20), gentamicin (0.5/5), and with sucrose at 3% (w/v) for cloning procedures. For microscopy analysis, cells were grown either from a single colony or from frozen stock. Serial dilutions (1:10, 1:50, 1:100, and 1:1000) were made, and cultures were grown overnight at 26 °C or 30 °C with shaking at 220 rpm before being imaged in mid-exponential phase. *E. coli* strains used in this study were grown in liquid lysogeny broth (LB) medium at 37 °C supplemented with antibiotics or supplements as necessary (diaminopimelic acid (DAP) 300 µg ml$^{-1}$, kanamycin 50 µg ml$^{-1}$, spectinomycin 100 µg ml$^{-1}$, gentamicin 15 µg ml$^{-1}$ and streptomycin 30 µg ml$^{-1}$). Strains were maintained on LB plates at 37 °C supplemented with antibiotics as necessary (kanamycin 50 µg ml$^{-1}$, spectinomycin 100 µg ml$^{-1}$, and streptomycin 30 µg ml$^{-1}$).

### Plasmid constructions and cloning procedures

All plasmids used in this study were cloned using standard molecular biology techniques and are listed in Supplementary Table S2. PCRs were performed using *A. excentricus* CB48 WT or mutant genomic DNA as template. Gibson assemblies were performed using the Gibson Assembly® Master Mix from NEB[65]. Sequences of the primers used are available in Supplementary Table S3.

In-frame deletions and fluorescent fusions in *A. excentricus* were obtained by double homologous recombination and sucrose counterselection, as previously described[66]. For deletions, 700-bp fragments from the upstream and downstream regions of the gene to be deleted were amplified by PCR. For the N-terminal mCherry fusion to PBP2, 500-bp of the upstream and N-terminus regions of the *pbp2* gene were amplified by PCR, along with the *mCherry* gene. PCR fragments were gel-purified and cloned using Gibson assembly into the suicide vector pNPTS139 that had been digested by *EcoRI* and *HindIII*. The pNPTS139-based constructs were transformed into *E. coli* DH5α cells, verified by PCR sequencing, and then introduced into *A. excentricus* via biparental mating using the dap⁻ *E. coli* strain WM3064 (YB7351)[67]. The pGFPC-5 plasmid with *egfp* replaced by *sfgfp* was used for generating the C-terminal sfGFP fusion to ZapA in *A. excentricus*. Proper chromosomal integration or gene replacement was verified by colony PCR and Sanger sequencing.

### Fluorescent D-amino acids (FDAAs)

In this study, three different FDAAs were used: HADA (7-hydroxycoumarin-3-carboxylic acid-D-alanine; emission peak ~ 407 nm), BADA (BODIPY FL-D-alanine; emission peak ~ 515 nm), and TADA (TAMRA-D-alanine; emission peak ~ 578 nm). FDAA stock solutions were prepared in anhydrous DMSO at a concentration of 100 mM. In fluorescent images and demographs, the FDAAs are false colored in green or magenta depending on the experiment, as detailed in the figure legends.

### FDAA pulse-chase experiments

To label whole cells, 250 µM TADA or BADA was added to early exponential phase cells (OD600 ~ 0.1). The cells were allowed to grow for two generations, after which they were washed three times with appropriate medium to remove excess FDAA from the medium.

Subsequently, growth was monitored following the wash using time-lapse microscopy, or cells were allowed to grow for half their generation times and imaged using phase and fluorescence microscopy to generate demographs (see "Image analysis" below).

For the FDAA pulse-chase experiments with mecillinam treatment, bacterial cells (OD600 ~ 0.1) were incubated with 250 µM TADA over two generations. Excess FDAA was removed by centrifugation at 6000 g for 3 min, and the cells were washed three times with PYE. Cell pellets were then resuspended in PYE with or without 50 µg ml$^{-1}$ mecillinam, grown for one additional doubling time, and imaged using phase and fluorescence microscopy.

### Dual short-pulse labeling with FDAAs

For dual short-pulse FDAA labeling in *C. crescentus*, HADA was added to early exponential phase cells (OD600 ~ 0.25) to a final concentration of 1 mM. The cells were grown in PYE at 30 °C for 5% of their doubling time (5 min). Then, the excess dye was removed by centrifugation at 6000 g for 3 min, and cells were washed 3 times with PYE. The cell pellets were resuspended in PYE, and the cells were allowed to grow for an additional 15% of their generation time (15 min) in fresh PYE. BADA was then added to the culture medium to a final concentration of 1 mM. The cells were grown for an additional 5% of their doubling time for sequential labeling. Excess BADA was then removed by centrifugation at 6000 × g for 3 min, and cells were washed 3 times with PYE. The labeled cells were resuspended in PYE and imaged with phase and fluorescence microscopy.

For dual short-pulse FDAA labeling in *A. excentricus*, TADA was added to early exponential phase cells (OD600 ~ 0.25) to a final concentration of 500 µM. The cells were grown in PYE at 26 °C for 5% of their doubling time (6 min). Then, excess TADA was removed by centrifugation at 6000 × g for 3 min, and cells were washed 3 times with PYE. The cell pellets were resuspended in PYE, and the cells were allowed to grow for an additional 15% of their generation time (18 min) in fresh PYE at 26 °C. BADA was then added to the culture medium to a final concentration of 500 µM. The cells were then grown for an additional 5% of their doubling time for sequential labeling. Excess BADA was removed by centrifugation at 6000 × g for 3 min, and cells were fixed in ethanol 70% for 1 h. The fixed cells were washed with PYE twice and imaged with phase and fluorescence microscopy.

For dual short-pulse FDAA labeling in *A. biprosthecum*, TADA was added to early exponential phase cells (OD600 ~ 0.25) to a final concentration of 250 µM. The cells were grown in PYE at 26 °C for 5% of their doubling time (7 min). Then, excess TADA was removed by centrifugation at 6000 × g for 3 min, and cells were washed 3 times with PYE. The cell pellets were resuspended in PYE, and the cells were allowed to grow for an additional 15% of their generation time (21 min) in fresh PYE at 26 °C. BADA was then added to the culture medium to a final concentration of 250 µM. The cells were then grown for an additional 5% of their doubling time for sequential labeling. Excess BADA was removed by centrifugation at 6000 × g for 3 min, and cells were fixed in ethanol 70% for 1 h. The fixed cells were washed with PYE twice and imaged with phase and fluorescence microscopy.

For dual short-pulse FDAA labeling in *R. capsulatus*, TADA was added to early exponential phase cells (OD600 ~ 0.25) to a final concentration of 250 µM. The cells were grown in PYS at 30 °C for 5% of their doubling time (6 min). Then, excess TADA was removed by centrifugation at 6000 × g for 3 min, and cells were washed 3 times with PYS. The cell pellets were resuspended in PYS, and the cells were allowed to grow for an additional 15% of their generation time (18 min) in fresh PYS at 26 °C. BADA was then added to the culture medium to a final concentration of 250 µM. The cells were then grown for an additional 5% of their doubling time for dual sequential labeling. Excess BADA was removed by centrifugation at 6000 g for 3 min, and cells were fixed in ethanol 70% for 1 h. The fixed cells were washed with PYS twice and imaged with phase and fluorescence microscopy

To orientate *A. excentricus* and *A. biprosthecum* cells using the old pole, holdfasts were detected with CF®405S conjugated wheat germ agglutinin (CF®405S WGA, 0.5 µg ml⁻¹ final concentration) since WGA binds specifically to the acetylglucosamine residues present in their holdfasts[68].

For FDAA dual short-pulse labeling in the presence of mecillinam, *A. excentricus* cells (OD$_{600}$ ~ 0.25) were treated with mecillinam (50 µg ml⁻¹) for 120 min. The treated cells were labeled with FDAAs as described above, but in the presence of 50 µg ml⁻¹ mecillinam for two sequential pulses of 5 min.

### Single short-pulse labeling with FDAA in *A. excentricus* and *C. crescentus* PBP2 fluorescent fusion cells

BADA and TADA were added to the early exponential phase (OD600 ~ 0.25) *C. crescentus gfp-pbp2* and *A. excentricus mCherry-pbp2* cells, to a final concentration of 250 µM. The *C. crescentus* and *A. excentricus* cells were grown in PYE at 30 °C and at 26 °C, respectively, for 5% of their doubling time. Excess FDAA was removed by centrifugation at 6000 × g for 3 min and cells were fixed with 70% ethanol for 1 h. Cells were washed 2 times with PYE, and labeled cells were then resuspended in PYE and imaged with phase and fluorescence microscopy.

### Microscopy

For light microscopy analysis, 24 mm × 50 mm coverslips (#1.5) were used as imaging supports for an inverted microscopy system. An agarose pad was made of 1% SeaKem LE Agarose (Lonza, Cat. No. 50000) in dH2O. Cell samples were loaded onto the coverslips. Then, an 8 mm × 8 mm × 2 mm (length, width, thickness) dH2O-agar pad was laid on top of the cells. The coverslip–pad combination was placed onto a customized slide holder on microscopes with the pad facing upwards.

For time-lapse, 1 µl FDAA-labeled cells were spotted onto pads made of 0.7% Gelrite (Research Product International, CAS. No. 71010-52-1) in PYE for *C. crescentus, A. excentricus,* and *A. biprosthecum* cells and topped with a glass coverslip. 1 µl FDAA-labeled *R. capsulatus* cells were spotted onto pads made of 0.7% Gelrite in PYS and topped with a glass coverslip. The coverslip was sealed with VALAP (vaseline, lanolin, and paraffin at a 1:1:1 ratio).

Images were recorded with inverted Nikon Ti-E or Ti2 microscopes using a Plan Apo 60 × 1.40 NA oil Ph3 DM objective with DAPI/FITC/Cy3/Cy5 or CFP/YFP/mCherry filter cubes and a Photometrics Prime 95B sCMOS camera. Images were processed with the NIS Elements software (Nikon).

### Image analysis

Cell dimensions were obtained using FIJI[69] and the plugin MicrobeJ[39]. To quantitatively analyze the pattern of FDAA loss in pulse-chase experiments, or of FDAA incorporation during dual short-pulse experiments, we generated kymographs and/or demographs using the MicrobeJ results interface. In these demographs, each cell is oriented such that the pole with the maximum mean fluorescence of the second FDAA is set to the right, and cells are aligned at the midcell. Alternatively, where holdfast staining was used, the holdfast signal (old pole) was set to the left. All cells were aligned at the midcell. Using the MicrobeJ demograph tool, 50% of the maximum intensity of each pulse was displayed in the main figures, while the total fluorescence signals are presented in the extended data. Demographs of the pulse-chase experiments of the different *Caulobacteraceae* species show 80% of the maximum fluorescence intensity. For all short-pulse analyses, cells undergoing division with apparent septation were excluded from the analysis, as were cells lacking a visible holdfast when holdfast staining was used to orient polarity.

To complement kymograph and demograph analysis, we generated fluorescence intensity plots showing the fluorescence signals from both FDAAs using MicrobeJ. For pulse-chase experiments, fluorescence intensities were measured in 25 cells at $t = 35$ min. Points along the medial axis of each cell were selected, and the FDAA signal was plotted relative to its normalized position along the cell length. For dual pulse-chase experiments, 50 cells were analyzed similarly, with normalized signals of FDAA 1 (magenta) and FDAA 2 (green) plotted along the cell length. Fluorescence intensity plots were generated using GraphPad Prism (version 10.3.0).

Subcellular localization heatmaps and density maps for mCherry-PBP2 and FDAA foci were generated using MicrobeJ using the "Maxima" detection option. Density map merges were produced by importing each density map into Adobe Illustrator CC 2023 (Adobe Inc.) and manually merging the plots.

To quantitatively analyze the patterns of FDAA loss and cell bulging during pulse-chase experiments with mecillinam treatment, we quantified the subcellular localization of bulges along with the fluorescence intensities of ZapA and TADA. Localization of the bulge was determined using the "feature" option of MicrobeJ. Instead of looking for constriction, we looked for bulging, using the option "inverted" in the feature parameters interface. Using the subcellular localization charts function of the MicrobeJ results interface, we plotted bulge distribution relative to the cell center and generated a ShapePlot based on cell length to localize ZapA fluorescence intensity and FDAA signal loss as a readout for PG synthesis. GraphPad Prism (v. 10.3.0) was used to generate histograms and fluorescence intensity profiles and to perform statistical analysis.

### In vitro mecillinam titration against PBPs

In vitro mecillinam titration against PBPs was performed with modifications to a previous protocol for PBP detection in *E. coli*[70]. Specifically, *A. excentricus* cells in exponential phase (OD$_{600}$ ~ 0.5) were harvested by centrifugation at 10,000 × g for 4 min at room temperature. The cell pellets were washed twice with 1 ml PBS (pH 7.4). Cells were then resuspended in 50 µl PBS containing 1, 10, or 100 µg ml⁻¹ of the antibiotic, while a reference sample was resuspended in 50 µl PBS without antibiotics. After 4 h of incubation at room temperature, cells were pelleted, washed with PBS, and resuspended in 50 µl PBS containing 5 µg ml⁻¹ Boc-FL. Following a 30-min incubation at room temperature, cells were pelleted, washed with 1 ml PBS, and then resuspended in 100 µl PBS. The cells were sonicated on ice using a Branson Sonifier 250 instrument (power setting 60 A, 30 s cycle for three 10 s intervals with 10 s of cooling time between rounds) to isolate the membrane proteome. The membrane pellet was then resuspended in 100 µl PBS and homogenized by sonication (power setting 20 A for 1 s). The protein concentration was measured using a NanoDrop 1000 Spectrophotometer and adjusted to 2.5 mg ml⁻¹ using PBS. Proteome samples (20 µl) were dispensed into clean 1.5 ml microcentrifuge tubes, and 10 µl of 2 × SDS-PAGE loading buffer was added to each sample. The samples were heated for 5 min at 90 °C to denature the proteins, cooled to room temperature, and then 25 µl of each sample was loaded onto a 4–15% SDS-PAGE precast gel. The gel was rinsed with distilled water three times and scanned using a Gel Doc XR system (Bio-Rad Laboratories, Inc) with a 526 nm short-pass filter.

### Immunoblot analysis

For immunoblot analysis, *A. excentricus* WT and mCherry-PBP2 expressing cells were grown to an OD$_{600}$ of ~ 0.3 and harvested by centrifugation. Whole-cell protein extracts were prepared, separated by SDS-PAGE, and transferred onto nitrocellulose membranes (Bio-Rad). Membranes were blocked for 1 h at room temperature in TBS containing 0.1% Tween 20 and 5% (w/v) non-fat dry milk, then incubated overnight at 4 °C with a polyclonal anti-mRFP antibody[71] (1:2500 dilution in TBS-T with 5% milk). Following incubation, membranes were washed four times for 5 min, then incubated for 1 h at room temperature with an HRP-conjugated goat anti-rabbit secondary

antibody (Pierce) diluted 1:10,000. Signal detection was performed using the SuperSignal™ West Pico Chemiluminescent Substrate (Thermo Fisher Scientific), and images were acquired with a Bio-Rad ChemiDoc™ system. Signal visualization and analysis were done using Image Lab software (version 6.0.1, Bio-Rad).

## Phylogenetic analysis of selected Alphaproteobacteria

Whole-genome data were obtained from the genome database maintained by the National Center for Biotechnology Information[72]. From each genome, a set of 37 conserved genes was identified and the translated amino acid sequences aligned and concatenated using Phylosift[73]. Phylogenetic reconstruction with MrBayes[74] used a mixed amino acid model including a four-category approximation of gamma-distributed rate variation and an invariant site category. Two simultaneous Markov chain runs were performed for 3,000,000 generations, discarding the initial 25% for burn-in. The tree was visualized and formatted using iTOL[75].

## Bioinformatic analyses and the phylogeny of PBPs

Identification of putative PBPs in *A. excentricus* was performed by BLAST analysis using *C. crescentus* PBP2 and PBP1a as queries. Amino acid sequences of homologues (including from other model organisms) were collected from UniProt (https://www.uniprot.org): PBP5 (BSU00100) in *B. subtilis*, DacD (JW5329), PBP1a (JW3359), PBP1b (JW0145), PBP1c (JW2503), PBP2 (JW0630), PBP3 (JW0082) and MTG (JW3175) in *E. coli*, PBP3 (Astex_1844), PBP2 (Astex_1631), PBP1a (Astex_2994), PBP1c (Astex_0196), PBP1b (Astex_2378) and MTG (Astex_0406) in *A. excentricus*, and PBP3 (CCNA_02643), PBP2 (CCNA_01615), PBPz (CCNA_93685), PBPc (CCNA_03386), PBPx (CCNA_01584), PBP1a (CCNA_01584) and MTG (CCNA_00328) in *C. crescentus*. Sequences were then aligned using MUSCLE v.3.8.31 (Fig. S5.1), and PhyML 3.2 was used to reconstruct the maximum likelihood phylogenetic tree, with automatic model selection by Smart Model Selection (SMS)[76] and Akaike information criterion. Phylogenetic reconstruction was performed by RAxML version 8.2.10[77] with 100 rapid bootstrap replicates to assess node support. The tree was visualized and formatted using iTOL[75]. Taxonomic assignments were based on the taxonomy database maintained by NCBI.

## Genomic organization of the *pbp2* and *mreB* genes

The genomic organization of the *pbp2* and *mreB* operons was analyzed using a combination of annotation, sequence search, and synteny visualization tools. Genome assemblies were first annotated with prokka[78]. The annotated files were organized into folders according to their formats (FAA, FNA, GFF, and GBK). For each FAA file, a BLAST protein database was created using the makeblastdb function, and the FNA files were indexed using samtools. GBK files were separated by contig using a Python script with BioPython's SeqIO library[79] to save each contig as an individual GBK file.

When protein files contained multiple sequences, they were split into individual files using SeqKit[80]. Each protein sequence was then identified by performing a BLAST search against the FAA databases. The blast search results were filtered to retain the first hit. Locus tags from the BLAST hits were extracted from the GFF files and converted to BED format. For analyses requiring extended genomic regions, the coordinates were expanded by 5000 bp upstream and downstream using bedtools slop.

To conduct synteny analysis, locus tags from the BED regions were extracted, and the corresponding protein sequences were retrieved from the FAA files using SeqKit[80]. A BLAST database was generated from these sequences, followed by an all-vs-all BLAST search. The matches were clustered based on sequence identity using the MCL algorithm[81], grouping homologous sequences.

Clusters were assigned specific colors, and the GBK files were edited by adding these colors for each corresponding locus tag.

Synteny diagrams were created with EasyFig[82], with coding regions represented as arrows and other genomic features (e.g., tRNA and rRNA genes) depicted as rectangles. Multiple synteny figures were arranged in the desired order and merged using Adobe Illustrator CC 2023 (Adobe Inc.).

## Nanopore sequencing methods

DNA samples were sequenced using Oxford Nanopore Technology. Library preparation and sequencing were conducted at the Bacterial Symbiont Evolution Lab (INRS, Laval, Canada) following the SQK-LSK109 kit protocol, with barcoding performed using the EXP-NBD114 kit. Briefly, 400 ng of DNA underwent end repair for barcode adapter ligation, followed by purification and pooling.

The pooled DNA was then used for adapter ligation, subjected to an additional purification step, and loaded onto a FLO-MIN106 flow cell for sequencing on the MinION Mk1C using MinKNOW (v21.05.12). Sequencing was performed until a minimum coverage of $100 \times$ per genome was achieved.

Basecalling was carried out with Guppy v6.1.1 using the guppy_basecaller sup model. Reads were demultiplexed, adapters were removed, and sequences with a quality score below 8 were filtered out using guppy_barcoder.

Assembly was performed using Canu v2.2 (https://github.com/marbl/canu), Flye v2.9.1 (https://github.com/fenderglass/Flye)[83], and Miniasm v0.3 (https://github.com/lh3/miniasm)[84]. The assemblies were refined using Trycycler v0.5.3 (https://github.com/rrwick/Trycycler) to generate a consensus sequence. The final assembly underwent base correction with Medaka v1.6.0 (https://github.com/nanoporetech/medaka).

Quality assessment of the assembly was performed using Quast v5.2.0 (https://github.com/ablab/quast)[85] and BUSCO v5.4.0 (https://gitlab.com/ezlab/busco)[86].

## Illumina sequencing methods

Illumina sequencing was performed by SeqCenter in Pittsburgh, PA. Sample libraries were prepared using the Illumina DNA Prep kit and IDT 10 bp UDI indices, and sequenced on an Illumina NextSeq 2000, producing $2 \times 151$bp reads. Demultiplexing, quality control and adapter trimming were performed with bcl-convert (v3.9.30).

## Variant calling methods

Illumina-generated $2 \times 151$bp paired-end read data was used as the input for variant calling against the provided GenBank AC48 reference. Variant calling was carried out using Breseq (v0.37.1) under default settings[87]. Mutations were confirmed by Sanger sequencing.

## Statistics and reproducibility

All experiments were independently repeated at least three times with similar results. No data were excluded from analyses unless otherwise specified (cells undergoing division, lacking holdfast staining, or appearing clumped). Statistical analyses were performed using GraphPad Prism 10 (Version 10.3.0 (461)), and differences between datasets were evaluated using unpaired two-sided Welch's $t$ tests. Protein localization patterns were assessed in at least three independent experiments with consistent outcomes. For imaging quantification, multiple images were analyzed per condition. Image fields and cells were selected at random for analysis.

## Reporting summary

Further information on research design is available in the Nature Portfolio Reporting Summary linked to this article.

# Data availability

The raw sequencing reads generated in this study have been deposited in the NCBI Sequence Read Archive (SRA) under accession codes

PRJNA1231080 [https://www.ncbi.nlm.nih.gov/sra/SRX27867301] and [https://www.ncbi.nlm.nih.gov/sra/SRX27867300]. The whole genome sequencing and variant calling analysis of *A. excentricus* mch-pbp2 generated in this study have been deposited in Figshare at https://doi.org/10.6084/m9.figshare.27623046. Source data are provided in this paper.

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

## Acknowledgements

The NIH supported this work with grants R35GM122556 to Y.V.B. and R35GM136365 to M.S.V. Y.V.B. is also supported by a Canada 150 Research Chair in Bacterial Cell Biology. M.D. was in part supported by a postdoctoral fellowship from the Swiss National Science Foundation (project #P2GEP3_191489, and P500PB_206676). F.J.V. received a Junior 1 and Junior 2 research scholar salary award from the Fonds de Recherche du Québec-Santé. Thanks to Yen-pang Hsu for synthesizing the FDAA. We thank Adrien Ducret for help with MicrobeJ. We thank Patrick Viollier for the antibodies. Many thanks to previous and current Brun Lab members for either providing guidance to the project or proofreading the manuscript.

## Author contributions

M.D., L.Y. and Y.V.B. designed the research. M.D. and L.Y. performed all the experiments, except for the whole-genome sequencing of the fluorescent fusion to *pbp2* strain performed by M.J. and whole-genome sequencing and genome assembly of *A. biprosthecum* and *P. conjunctum* performed by F.P. The phylogenetic analysis of Alphaproteobacteria species was conducted by D.T.K., while M.D. carried out the bioinformatic analyses and, together with K.A.G., performed the phylogeny of PBPs. M.S.V. contributed fluorescent D-amino acids (FDAAs). M.D., L.Y., V.H. and Y.V.B. analyzed the overall data. M.D., L.Y., V.H. and Y.V.B. wrote the manuscript. M.D., F.J.V. and Y.V.B. acquired funding.

## Competing interests

The authors declare no competing interests.
