## [Transparent Peer Review file · Nature Communications]

Phenotypic plasticity in cell elongation among closely related bacterial species

Corresponding Author: Dr Yves Brun

Version 0:

Reviewer comments:

Reviewer #1

(Remarks to the Author)

The manuscript offers an insightful investigation into bacterial cell elongation, a fundamental process in the bacterial cell cycle and a key antibiotic target. By addressing the limitations of research on a narrow range of model species, the authors provide valuable perspectives on the diversity and adaptability of elongation mechanisms, supported by clear data and well-designed figures.

Using fluorescent D-amino acids (FDAAs), the study reveals significant variation in elongation strategies among Caulobacteraceae species, encompassing dispersed, midcell, and polar elongation with both unidirectional and bidirectional dynamics. Genetic, cell biology, and phylogenetic analyses link unidirectional midcell elongation to changes in the localization of peptidoglycan synthase PBP2.

This work emphasizes the phenotypic plasticity of elongation complexes within Caulobacteraceae and across Alphaproteobacteria, challenging existing paradigms and advancing our understanding of bacterial growth, morphology, adaptation, and antibiotic resistance.

Questions/points

1. The observed differences in elongation strategies among Caulobacteraceae species are intriguing. However, could these differences be influenced by how cells perceive or adapt to specific growth conditions in the experimental setup? For example, do nutrient availability, osmotic stress, or other environmental factors impact the localization or activity of elongosome components such as PBP2? In other words, how robust are the localization patterns?
2. The data in Fig. 1D, Fig. 5, and Ext. Data Fig. 1 clearly show bidirectional elongation in *C. crescentus*. However, the figures give the impression that there may be a slight preference for extension on one side of the future division site. For example, in Fig. 5, the "left side" of the dotted line appears to have more pronounced blackness compared to the right. Could the authors clarify whether this asymmetry is meaningful or simply an artifact of the visualization?
3. It would be valuable to discuss and extrapolate the described findings in the context of polar growth, which is similarly widespread across bacterial systems. A particularly intriguing comparison is with DivIVA, a key protein in polar growth in Actinobacteria, which plays distinctly different roles in other bacteria, such as *Bacillus subtilis*. Could the mechanisms and insights presented in this study shed light on the functional versatility of such conserved proteins, or are these systems too fundamentally different to draw meaningful parallels? Exploring this connection in the discussion might provide a broader perspective on the evolutionary adaptability of bacterial growth strategies.

Reviewer #2

(Remarks to the Author)

This manuscript describes studies on the diversity of elongation modes in closely related species. A first major difference was found between the elongation mode of *A. excentricus* and *C. crescentus*. While the latter elongates bidirectionally from the midcell to both poles, *A. excentricus* elongates only unidirectionally from the midcell towards the new pole only. They then provide molecular insights into these two different elongation modes by showing that PBP2 activity and localization are critical for the unidirectional elongation at the midcell in *A. excentricus*. Finally, they provide evidence that these two

elongation modes are found in other Caulobacteraceae. However, and very interestingly, these analyses also revealed that some species (and in particular *A. biprosthecum*) have, in addition to the unidirectional elongation from the midcell to the new pole, another elongation mode located at the pole itself. Studies of *R. capsulatus* as a representant of Rhodobacterales (distinct from Caulobacteraceae) further suggest that the unidirectional elongation mode may be widespread within the alphaproteobacterial clade.

This is a very well written manuscript describing a series of carefully conducted experiments. The conclusions are supported by the experiments made. I think the results presented will be of great interest to microbiologists, especially those studying the bacterial cell morphogenesis. There is little room for criticism and I have only the following comments to improve the manuscript:

- line 56: mention the SEDS as they are also required for PG synthesis.

- lines 58-64: Consider to cite MreB, FtsZ and DivIVA as specific scaffolds for dispersed, localized and polar cell elongation. Regarding the later, reference could be made to the "polarisome" described in the following two papers (Flardh et al, Curr Opin Microbiol, 2012 and Hempel et al, Proc Natl Acad USA, 2012).

- Figure 1b: according to fig1a, the polar red arrowheads in *A. biprosthecum* could be reversed.

- Introduction: I agree that with the authors' statements that the different cell elongation modes lead to diverse cell shapes and conversely, that the same elongation mode can lead to a diversity of cell shapes. However, it is not clear to me why the authors hypothesize that elongation strategies might be different in closely related species. Can they provide more details?

- line 81: to help the reader, specify that PBP2 is a class B PBP.

- line 90: closely related species... but only if they have different cell shape? Do the authors think that closely related species with the same morphological features could have different elongation modes?

-lines 118-121: I agree that these conclusions are consistent with what was observed in the cells presented in Fig1d and ExtDataFig1. However, a plot (normalized cell length as a function of mean fluorescence intensity (like in Fig 2B)) generated with the fluorescence signal from many cells would strengthen this conclusion. On the other hand, it is less obvious that the fluorescence signal decreases at the old pole of *A. excentricus*. I suggest quantifying the fluorescence signal of the old pole once cell constriction is initiated to show that it does indeed decrease while that of the new pole remains stable.

- line 124-125: When I read these lines, I thought about two experiments to support the conclusions. However, the data then presented later in the manuscript are convincing and I think that these experiments is not really necessary. I just mention them below so that the authors can consider them, especially if they have already done these some of them, and include them in the manuscript to strengthen their conclusion:

It would be worth measuring the distances between the old pole and ZapA and the new pole and ZapA at the start of cell constriction and until the end of cell elongation. The idea is to determine whether PG synthesis occurs at the same rate on both sides during septation and constriction. It is expected that the elongation rate of the stalked cell will be lower than that of the swarmer cell. This will confirm that cell elongation is unidirectional and not bidirectional with different kinetics. Second, short pulse labeling with FDAA should show that new PG is synthesized in the stalked cell only after the initiation of cell septation and constriction.

- line 155: as for *C. crescentus*, it might be nice to show FDAA signals plotted against their relative position along the cell for a subset of cells.

- line 187: to be fully conclusive about the offset localization towards the new pole, I propose to localize PBP2 together with ZapA-sfGFP.

- lines 189-192: Did the author try to perform SIM? This would provide better support for this conclusion.

-line 213: I think there is a sentence missing to say that *A. excentricus* cells also bulge in the presence of mecillinam.

- line 122-123 and elsewhere: Perhaps these method details could be removed. They are present in the M&M and there are schematics in each figure (1a, 2a, 4a, 5a and 6a), which are well described by the corresponding legend.

- line 235: as in Fig 1d, a kymograph might be helpful to give a nice visual illustration of shapeplots.

- line 275: This observation implies that the PBP2 homolog should be localized at the pole. Is it possible to test this in *A. biprosthecum*?

-line 281 : why mainly at the pole? It should also be intense in the swarmer cells near midcell. Can the authors comment on this. Is it due to technical issues or is it biologically relevant?

Fig 6c: a red FDAA band is missing to the left side of the green FDAA in *A. biprosthecum* (as in ExtDataFig 6c) and the polar green is not very obvious (better in ExtDataFig 6c).

- line 289: I'm just curious, but how did the authors choose *R. capsulatus* to represent Rhodobacterales?
- line 367-376: please comment on how PBP2 localization might be modulated. What might be the mechanism?

Reviewer #3

(Remarks to the Author)

Delaby et al., investigate the mechanism of cell elongation using fluorescent D-amino acids (FDAAs) in closely related Caulobacteraceae. Performing pulse-chase experiments in *Caluobacter crescentus* and *Asticcacaulis excentricus* they find striking differences in the FDAA insertion pattern. While *C. crescentus* elongates from bi-directionally from mid-cell, *A. excentricus* elongates asymmetrically towards the new pole. This suggests that despite their close phylogenetic relationship these organisms employ distinct mechanism of zonal cell elongation, either bidirectional towards the old and new pole or unidirectional towards the new pole, respectively. This finding is further supported based on the localization pattern of an elongasome associated cell wall synthase, PBP2. While in *C. crescentus* PBP2 is uniformly distributed in the cell membrane and does not show specific colocalization with short FDAA pulses, the *A. excentricus* homologue co-localizes towards zones of new cell wall incorporation at mid-cell. Additionally, cell-elongation is dependent on PBP2 as shown through inhibition by mecillinam. Last, the authors further show that uni-directional and bi-directional cell elongation arose multiple time independently during the evolution and are even present in other Alphaproteobacteria species (*Rhodobacter capsulatus*), while identifying a third type of zonal elongation in *A. biprosthecum* displaying a mixture polar and unidirectional midcell elongation.

Overall, this is an extremely intriguing finding and serves a textbook example showcasing the complex diversity of bacterial morphogenesis. Unfortunately, this is abstracted all too often by drawing premature conclusion based on molecular mechanism of a few model organisms and extrapolated by phylogenetic analyses. Yet, cytological assays as presented here are often lacking, thus significantly masking our understanding of bacterial cell biology. However, there are a few critical points that should be addressed prior to publication.

Major comments:

- Throughout the study FDAAs (HADA, BADA and TADA) are used as to visualize PG remodeling. While HADA has been used extensively to label PG in Gram-negative bacteria, I am concerned about the outer membrane (OM) permeability of BADA and TADA (PMID: 25474031). In Figure 5a it appears as if hold-fast of *A. biprosthecum*, *C. henricii* and *P. conjunctum* are stained, which to my knowledge should not contain PG, at least under non-starved conditions (PMID: 18757530, PMID: 30707707). Furthermore, often it is hard to discern a specific surface localization of the staining of these dyes in *A. excentricus* (e.g. Figures 2c,d; 3b,c; 4a,c; 5d,g), while in *C. crescentus* (Fig 1b) the staining is beautiful. Why do the authors not use the same FDAAs for staining *A. excentricus*? Since these experiments are the center piece of this study, I suggest that the authors perform a control experiment performing FDAA labeling while subsequently isolating PG sacculi to demonstrate that the observed staining is indeed associated with the cell wall and not the result of excess dye sticking to the cell surface. This should be done for all species which have not be previously stained with FDAAs. Alternatively, the authors could also consider performing labeling experiments with biorthogonal D-Ala-D-Ala peptides, which would more directly report on de novo PG synthesis (PMID: 24336210).

- The authors suggest that differentially regulated elongasome activity (e.g. PBP2 localization) is the underlying reason for the distinct cell elongation patterns. Unfortunately, they provide only very limited experimental support and mechanistic insights to this end. To solidify this point I encourage the authors to perform additional experiments. Previously, A22 has been used to inhibited MreB polymerization in *C. crescentus* (PMID: 22505677). Adding such a piece of data would strongly support the mecillinam findings. More generally, localization of the cytoskeletal network (MreB) or further elongasome components (RodA) would buttress the differences in PBP2 localization. Fluorescent fusion to these protein have been successfully engineered in *C. crescentus* (PMID: 22505677, PMID: 30707707). Genetic or biochemical data are lacking altogether. Is RodA-PBP2 essential in *A. excentricus* or can PBP2 interact with alternative GT-enzymes. Generating active site point mutations in RodA or PBP2 as well as elongasome depletion strains (RodA, PBP2, MreB, MreC, MreD) and comparing their terminal phenotype could further support the authors hypothesis. Alternatively, B2H screens could be performed to demonstrate interaction of the elongasome components. Last, additional bioinformatic analysis such as amino acid sequence comparison or AlphaFold could reveal structural differences or unique domains between the different elongasome components which might point out sites for alternative regulation or conserved residues.

Minor comments:

- When zooming in on micrographs it seems that all microscopy images have been interpolated in the pdf file. Please check if microscopy images are of sufficient quality.
- The red/green color scheme is not color blind friendly. Consider replacing with green/magenta or cyan/yellow.
- I would encourage the author to show more zoomed in views of single cells. While I appreciate to see multiple cells at once, it is unfortunately often hard to discern the staining pattern. Given that the authors generally show demographs and density plots with high N numbers, I think this would be totally fine and make it easier for the reader to see what the authors want to point out.
- Kymograph montages in Figure 1d, 6b,f; S1, S6b, S7b are all lacking a scale bar. Please include.

- All demographs and heatmaps lack a display of the intensity values in their lookup table. This is particularly important for non-linear LUTs such as 'Fire' (Fig. 3a). Moreover, while the heatmap and the micrograph show a clear enrichment at mid-cell for PBP2 in *A. excentricus*, this is far less obvious in the demograph. This seems to be based on the normalization of the signal. Consider showing raw intensities.

- PBP2 localization: Why do the authors use different fluorescent molecules to localize the same protein in different organisms? If possible, the authors should check whether swapping the fluorescent molecule does affect the staining pattern. Also since the author fix the cells in 70% EtOH (which denaturants proteins), I wonder if this might affect GFP and mCherry fusions to different degrees.

- I appreciate that the authors whole-genome-sequenced their mCh-PBP2 and report on the point mutations in FtsW, however this is not mentioned at any point in the main text or any reference is made to the additional text/figure. I would also encourage the authors to add phase-contrast images of wt vs mCh-PBP2 cells along with their quantifications (Figure S8a). Furthermore, WB should be provided to demonstrate the protein is expressed as a full-length fusion.

- LL251-253: As outlined above, I would encourage the authors to perform further experiments or tone this statement down.

- LL451: Primer sequence should be made available in a supplementary table.

- The authors should make their quantification more homogenous, e.g. cell length and width are once represented as dot blots (Figure S8a), and violin plots (Figure S8c), while fluorescence intensity is represented as bar graphs (Figure 4d). I suggest using dot plots (not black symbols) so that mean/median above the data points can be easily seen.

Version 1:

Reviewer comments:

Reviewer #3

(Remarks to the Author)

I have read the revised version of the article by Delaby and most of my comments were either fully addressed or declined with reasonable explanation. I thus recommend this article for publication although I still have a minor stylistic comment which should be addressed prior to copy editing:

The microscopy images in the manuscript are still interpolated. I checked the figshare link and have provided a screen shot of Fig. 3a to the editor (which he can share with you) to illustrate this. While the original micrographs (center, opened in FIJI) are fine (although WGA signal is saturated!), they appear interpolated in both the summarized reviewer pdf (left), as well as the separate SVG source file (right). The same is true for the bioRxiv version of this manuscript. This can be easily seen by the absence of discernable pixels in the image. Supplementary Fig. 8 is another clear example.

One easy way to circumvent this would be to scale the crops 4x4 (without interpolation) and subsequently saving as tiff files. This can easily be done in FIJI and will result in much nicer microscopy images without interpolation artefacts. Seeing the pixels at this magnification is a good sign!

While this doesn't change any of the conclusion of this paper, I just think it's a shame that a such beautiful story is hampered by less-than-ideal image rendering. This paper is going to be well received by the community, highly cited and will certainly make its way in many lectures, potentially textbooks. I would use the chance to correct this now.

Response to reviewer's comments:

Reviewer #1 (Remarks to the Author):

The manuscript offers an insightful investigation into bacterial cell elongation, a fundamental process in the bacterial cell cycle and a key antibiotic target. By addressing the limitations of research on a narrow range of model species, the authors provide valuable perspectives on the diversity and adaptability of elongation mechanisms, supported by clear data and well-designed figures.

Using fluorescent D-amino acids (FDAAs), the study reveals significant variation in elongation strategies among Caulobacteraceae species, encompassing dispersed, midcell, and polar elongation with both unidirectional and bidirectional dynamics. Genetic, cell biology, and phylogenetic analyses link unidirectional midcell elongation to changes in the localization of peptidoglycan synthase PBP2.

This work emphasizes the phenotypic plasticity of elongation complexes within Caulobacteraceae and across Alphaproteobacteria, challenging existing paradigms and advancing our understanding of bacterial growth, morphology, adaptation, and antibiotic resistance.

We thank the reviewer for these positive comments.

Questions/points

1. The observed differences in elongation strategies among Caulobacteraceae species are intriguing. However, could these differences be influenced by how cells perceive or adapt to specific growth conditions in the experimental setup? For example, do nutrient availability, osmotic stress, or other environmental factors impact the localization or activity of elongosome components such as PBP2? In other words, how robust are the localization patterns?

We thank the reviewer for this insightful question. We acknowledge that environmental conditions, such as nutrient availability or osmotic stress, could influence cell elongation strategies and the localization or activity of elongosome components. However, our study was designed to compare elongation mechanisms across species under standardized conditions, and systematically investigating the impact of environmental factors falls beyond the scope of this work. This remains an interesting avenue for future research.

2. The data in Fig. 1D, Fig. 5, and Ext. Data Fig. 1 clearly show bidirectional elongation in *C. crescentus*. However, the figures give the impression that there may be a slight preference for extension on one side of the future division site. For example, in Fig. 5, the "left side" of the dotted line appears to have more pronounced blackness compared to the right. Could the authors clarify whether this asymmetry is meaningful or simply an artifact of the visualization?

We thank the reviewer for this careful observation. While *C. crescentus* exhibits bidirectional elongation from the midcell, we consistently observe a slight asymmetry in labeling, with a seemingly more pronounced signal on the old pole side. This may be biologically meaningful and could stem from the inherent asymmetry in *Caulobacteraceae* division. Given that *C. crescentus* divides asymmetrically into a

swarmer and a stalked cell, the latter is typically longer at the time of labeling and may incorporate more PG during elongation. While this pattern is reproducible, systematically quantifying any potential bias in elongation direction would require additional investigation. To address this point, we have clarified the description of bidirectional elongation in the manuscript (line 153) to acknowledge the observed asymmetry and its possible explanations with the following text: “Notably, we also observed a slight asymmetry in fluorescence intensity, with a stronger signal on the old pole side. Given that *C. crescentus* divides asymmetrically into a swarmer and a stalked cell, and that the stalked cell is typically longer at the time of labeling, it may incorporate more PG during elongation. This could result in apparent differences in labeling while still maintaining bidirectional elongation.”

3. It would be valuable to discuss and extrapolate the described findings in the context of polar growth, which is similarly widespread across bacterial systems. A particularly intriguing comparison is with DivIVA, a key protein in polar growth in Actinobacteria, which plays distinctly different roles in other bacteria, such as *Bacillus subtilis*. Could the mechanisms and insights presented in this study shed light on the functional versatility of such conserved proteins, or are these systems too fundamentally different to draw meaningful parallels? Exploring this connection in the discussion might provide a broader perspective on the evolutionary adaptability of bacterial growth strategies.

We thank the reviewer for this interesting suggestion and agree on the relevance of discussing the broader theme of functional versatility in conserved proteins. Our findings highlight how bacterial elongation mechanisms can diverge significantly even among closely related species, raising the possibility that conserved proteins, such as PBP2 or DivIVA, have adapted to different organizational mechanisms depending on cellular and evolutionary contexts. To address this point, we have expanded our discussion (lines 402 to 411) to provide a broader perspective on the evolutionary adaptability of bacterial growth strategies, including a comparison to DivIVA mediated polar growth.

Reviewer #2 (Remarks to the Author):

This manuscript describes studies on the diversity of elongation modes in closely related species. A first major difference was found between the elongation mode of *A. excentricus* and *C. crescentus*. While the latter elongates bidirectionally from the midcell to both poles, *A. excentricus* elongates only unidirectionally from the midcell towards the new pole only. They then provide molecular insights into these two different elongation modes by showing that PBP2 activity and localization are critical for the unidirectional elongation at the midcell in *A. excentricus*. Finally, they provide evidence that these two elongation modes are found in other Caulobacteraceae. However, and very interestingly, these analyses also revealed that some species (and in particular *A. biprosthecum*) have, in addition to the unidirectional elongation from the midcell to the new pole, another elongation mode located at the pole itself. Studies of *R. capsulatus* as a representant of Rhodobacterales (distinct from Caulobacteraceae) further suggest that the unidirectional elongation mode may be widespread within the alphaproteobacterial clade.

This is a very well written manuscript describing a series of carefully conducted experiments. The conclusions are supported by the experiments made. I think the results presented will be of great interest to microbiologists, especially those studying the bacterial cell morphogenesis. There is little room for criticism and I have only the following comments to improve the manuscript:

We thank the reviewer for these positive comments.

- line 56: mention the SEDS as they are also required for PG synthesis.

We thank the reviewer for this suggestion. SEDS (Shape, Elongation, Division, and Sporulation) proteins are indeed essential for PG synthesis and function alongside PBP2 in elongation. To acknowledge their role, we have revised the text at line 56 to include SEDS proteins in the description of PG synthesis machinery.

- lines 58-64: Consider to cite MreB, FtsZ and DivIVA as specific scaffolds for dispersed, localized and polar cell elongation. Regarding the later, reference could be made to the “polarisome” described in the following two papers (Flardh et al, Curr Opin Microbiol, 2012 and Hempel et al, Proc Natl Acad USA, 2012).

We thank the reviewer for this suggestion. MreB, FtsZ, and DivIVA are indeed key scaffolds for dispersed, localized, and polar cell elongation, respectively. However, to keep the introduction concise and avoid adding too much detail at this stage, we have instead incorporated a discussion of their roles, including the role of DivIVA in polar elongation, in the discussion section line 383.

- Figure 1b: according to fig1a, the polar red arrowheads in *A. biprosthecum* could be reversed.

We thank the reviewer for this observation. We have now corrected this in Figure 1b to accurately reflect the elongation patterns described in Figure 1a.

- Introduction: I agree that with the authors' statements that the different cell elongation modes lead to diverse cell shapes and conversely, that the same elongation mode can lead

to a diversity of cell shapes. However, it is not clear to me why the authors hypothesize that elongation strategies might be different in closely related species. Can they provide more details? related species. Can they provide more details?

Historically, we did not hypothesize that the closely related species we study had different elongation mode. We simply noticed that *A. biprosthecum* elongated polarity and then looked in more details at *A. biprosthecum* and *A. excentricus* in comparison to *C. crescentus*. Even if we wanted to develop a hypothesis a posteriori to make us look smarter, it is not obvious that this can be done. Therefore, no changes have been made.

- line 81: to help the reader, specify that PBP2 is a class B PBP.

Done

- line 90: closely related species... but only if they have different cell shape? Do the authors think that closely related species with the same morphological features could have different elongation modes?

We think closely related species can have different elongation modes irrespective of their morphology.

-lines 118-121: I agree that these conclusions are consistent with what was observed in the cells presented in Fig1d and ExtDataFig1. However, a plot (normalized cell length as a function of mean fluorescence intensity (like in Fig 2B)) generated with the fluorescence signal from many cells would strengthen this conclusion. On the other hand, it is less obvious that the fluorescence signal decreases at the old pole of *A. excentricus*. I suggest quantifying the fluorescence signal of the old pole once cell constriction is initiated to show that it does indeed decrease while that of the new pole remains stable.

We thank the reviewer for this helpful suggestion. To strengthen this conclusion, we have generated plots of normalized cell length as a function of mean fluorescence intensity, similar to Fig. 2B, and included them in Extended Data Fig. 1b and 1d. Additionally, we have updated the results section (lines 123-126) to reference these new data.

Regarding the suggestion to quantify fluorescence at the old pole once cell constriction is initiated, we appreciate the reviewer's perspective. However, we believe there may be a slight misunderstanding. Our statement refers to the loss of FDAA signal on the old pole side of the division site (not the old pole itself) during constriction (Fig. 1d, white stars), which we interpret as septation-related PG synthesis. As our study focuses on cell elongation mechanisms rather than division, our analysis is centered on elongation patterns prior to division. For this reason, we have not conducted additional quantification in this context.

- line 124-125: When I read these lines, I thought about two experiments to support the conclusions. However, the data then presented later in the manuscript are convincing and I think that these experiments is not really necessary. I just mention them below so that the authors can consider them, especially if they have already done these some of them, and include them in the manuscript to strengthen their conclusion:

It would be worth measuring the distances between the old pole and ZapA and the new pole and ZapA at the start of cell constriction and until the end of cell elongation. The idea is to determine whether PG synthesis occurs at the same rate on both sides during

septation and constriction. It is expected that the elongation rate of the stalked cell will be lower than that of the swarmer cell. This will confirm that cell elongation is unidirectional and not bidirectional with different kinetics. Second, short pulse labeling with FDAA should show that new PG is synthesized in the stalked cell only after the initiation of cell septation and constriction.

We thank the reviewer for these insightful suggestions. We appreciate the proposed experiments and agree that they could further explore the dynamics of PG synthesis during septation and constriction. While our current study focuses on elongation rather than division, these approaches align with our future research directions to better understand the interplay between elongation and septation.

- line 155: as for *C. crescentus*, it might be nice to show FDAA signals plotted against their relative position along the cell for a subset of cells.

Done

- line 187: to be fully conclusive about the offset localization towards the new pole, I propose to localize PBP2 together with ZapA-sfGFP.

We thank the reviewer for this suggestion. However, given the resolution limits of fluorescence microscopy, we believe that co-localizing PBP2 with ZapA-sfGFP would not provide additional conclusive insights beyond our current data. Due to these limitations, the two signals might appear colocalized even if they are slightly offset in reality.

- lines 189-192: Did the author try to perform SIM? This would provide better support for this conclusion.

We appreciate the reviewer's suggestion and fully agree that SIM or other high-resolution microscopy approaches would provide valuable insights. In fact, as part of our ongoing research, we plan investigate PBP2 dynamics and PG synthesis using advanced microscopy techniques, including single-particle tracking and SIM.

-line 213: I think there is a sentence missing to say that *A. excentricus* cells also bulge in the presence of mecillinam.

We thank the reviewer and agree that a sentence is missing to state that *A. excentricus* cells also bulge in the presence of mecillinam improves clarity. We have now revised the text accordingly to clearly mention this observation (lines 224-225).

- line 122-123 and elsewhere: Perhaps these method details could be removed. They are present in the M&M and there are schematics in each figure (1a, 2a, 4a, 5a and 6a), which are well described by the corresponding legend.

We appreciate the reviewer's suggestion. However, we believe it is important to retain these methodological details in the main text, as the experimental approaches vary slightly across different analyses. Keeping these concise descriptions ensures clarity and helps the reader follow the results without having to refer back to the Methods section frequently.

- line 235: as in Fig 1d, a kymograph might be helpful to give a nice visual illustration of shapeplots.

We thank the reviewer for this suggestion. We assume the reviewer meant a demograph rather than a kymograph, as our experiment does not involve time-lapse imaging. Specifically, we labeled the cells with FDAA for two generations, washed out the excess, and then allowed them to grow in fresh media without FDAA for 2 hours before imaging them by fluorescence microscopy. Since this approach captures a population snapshot rather than continuous temporal dynamics in individual cells, a demograph is more appropriate for visualizing the results. We have now included demographs in Extended Data Figure 5c, along with a corresponding description in the text lines 246-248.

- line 275: This observation implies that the PBP2 homolog should be localized at the pole. Is it possible to test this in *A. biprosthhecum*?

We thank the reviewer for this insightful suggestion. To address this question, we attempted to tag *A. biprosthhecum* PBP2 using a merodiploid strain. However, even in the absence of induction, the fluorescent fusions caused significant morphology defects, making it challenging to assess PBP2 localization (see microscopy image below). While determining the precise localization of PBP2 in *A. biprosthhecum* is an interesting question, we believe it is beyond the scope of this study. Nevertheless, we recognize its potential relevance and plan to explore it further in future work.

-line 281: why mainly at the pole? It should also be intense in the swarmer cells near midcell. Can the authors comment on this. Is it due to technical issues or is it biologically relevant?

We thank the reviewer for this insightful comment. We think that polar growth may occur first, leading to a stronger FDAA signal at the pole. To clarify our analysis, we labeled cells with WGA and only retained those that were successfully labeled.

Additionally, cells that had initiated septation were excluded to prevent conflating midcell elongation with division-related processes.

Based on our experiments using both short and long FDAA pulses, we propose that swarmer cells primarily undergo dispersed elongation. In contrast, stalked cells initially elongate polarly before transitioning to midcell-directed unidirectional elongation, ultimately leading to division. However, because we exclude cells that have initiated division and exhibit septation, our analysis may introduce a bias by inadvertently omitting cells undergoing midcell elongation either during the first pulse or between the two labeling periods.

While we cannot entirely rule out technical limitations, we believe that the observed localization pattern of elongation is a biologically relevant aspect of *A. biprosthecum* growth dynamics.

Fig 6c: a red FDDA band is missing to the left side of the green FDAA in *A. biprosthecum* (as in ExtDataFig 6c) and the polar green is not very obvious (better in ExtDataFig 6c).

Corrected

- line 289: I'm just curious, but how did the authors choose *R. capsulatus* to represent Rhodobacterales?

We choose *R. capsulatus* because it's another model organism studied in our lab, particularly for polar adhesion. Given its phylogenetic placement within the *Rhodobacterales*, we were curious to explore its growth dynamics in comparison to the *Caulobacteraceae*.

- line 367-376: please comment on how PBP2 localization might be modulated. What might be the mechanism?

We appreciate the reviewer's suggestion and have incorporated a discussion on potential mechanisms regulating PBP2 localization. We now mention that PBP2 positioning in *A. excentricus* could be influenced by interactions with divisome or elongasome components, as well as specific features of the peptidoglycan itself. Additionally, we highlight recent findings in *E. coli* suggesting that PBP2 may act as a key driver of elongation, potentially responding to peptidoglycan structural cues to recruit the elongation machinery. This discussion has been added to lines 408-419.

Reviewer #3 (Remarks to the Author):

Delaby et al., investigate the mechanism of cell elongation using fluorescent D-amino acids (FDAAs) in closely related Caulobacteraceae. Performing pulse-chase experiments in *Caluobacter crescentus* and *Asticcacaulis excentricus* they find striking differences in the FDAA insertion pattern. While *C. crescentus* elongates from bi-directionally from mid-cell, *A. excentricus* elongates asymmetrically towards the new pole. This suggests that despite their close phylogenetic relationship these organisms employ distinct mechanism of zonal cell elongation, either bidirectional towards the old and new pole or unidirectional towards the new pole, respectively. This finding is further supported based on the localization pattern of an elongosome associated cell wall synthase, PBP2. While in *C. crescentus* PBP2 is uniformly distributed in the cell membrane and does not show specific colocalization with short FDAA pulses, the *A. excentricus* homologue co-localizes towards zones of new cell wall incorporation at mid-cell. Additionally, cell-elongation is dependent on PBP2 as shown through inhibition by mecillinam. Last, the authors further show that uni-directional and bi-directional cell elongation arose multiple time independently during the evolution and are even present in other Alphaproteobacteria species (*Rohdobacter capsulatus*), while identifying a third type of zonal elongation in *A. biprosthecum* displaying a mixture polar and unidirectional midcell elongation.

Overall, this is an extremely intriguing finding and serves a textbook example showcasing the complex diversity of bacterial morphogenesis. Unfortunately, this is abstracted all too often by drawing premature conclusion based on molecular mechanism of a few model organisms and extrapolated by phylogenetic analyses. Yet, cytological assays as presented here are often lacking, thus significantly masking our understanding of bacterial cell biology.

However, there are a few critical points that should be addressed prior to publication.

We thank the reviewer for these comments.

Major comments:

- Throughout the study FDAAs (HADA, BADA and TADA) are used as to visualize PG remodeling. While HADA has been used extensively to label PG in Gram-negative bacteria, I am concerned about the outer membrane (OM) permeability of BADA and TADA (PMID: 25474031). In Figure 5a it appears as if hold-fast of *A. biprosthecum*, *C. henricii* and *P. conjunctum* are stained, which to my knowledge should not contain PG, at least under non-starved conditions (PMID: 18757530, PMID: 30707707). Furthermore, often it is hard to discern a specific surface localization of the staining of these dyes in *A. excentricus* (e.g. Figures 2c,d; 3b,c; 4a,c; 5d,g), while in *C. crescentus* (Fig 1b) the staining is beautiful. Why do the authors not use the same FDAAs for staining *A. excentricus*? Since these experiments are the center piece of this study, I suggest that the authors perform a control experiment performing FDAA labeling while subsequently isolating PG sacculi to demonstrate that the observed staining is indeed associated with the cell wall and not the result of excess dye sticking to the cell surface. This should be done for all species which have not be previously stained with FDAAs.

Alternatively, the authors could also consider performing labeling experiments with biorthogonal D-Ala-D-Ala peptides, which would more directly report on de novo PG synthesis (PMID: 24336210).

We appreciate the reviewer's concern regarding FDAA labeling and outer membrane permeability. Indeed, BADA and TADA are larger molecules than HADA, but we opted to use them because HADA is less stable and prone to photobleaching, which affects imaging quality. Regarding Fig. 5a, the structures labeled in *A. biprosthicum*, *C. henricii*, and *P. conjunctum* are not holdfasts but stalks, which are made of PG (PMID: 10629178, PMID: 30707707). Regarding the choice of FDAAs for *A. excentricus*, we note that cells may tolerate different FDAAs to varying degrees. *Asticcacaulis* species grew well in the presence of BADA and TADA, which is why we used them in this study. To address the reviewer's concerns further, we performed the suggested control experiment, isolating PG sacculi following FDAA labeling in both *Asticcacaulis* species. This confirms that the observed staining is associated with the cell wall rather than excess dye adhering to the cell surface. While we did not include these results in the manuscript, we provide the corresponding figures below for reference.

Methods:

FDAA pulse-chase experiments and sacculi purification: Sacculi from cells were purified as described (Kuru *et al.*, 2012, PMID: 23055266) with the following modifications: to label whole *A. excentricus* and *A. biprosthicum* cells, 250 μ M TADA or BADA was added to early exponential phase cells (OD600 ~0.1). The cells were allowed to grow for two generations, after which they were washed two times with PYE to remove excess FDAA from the medium. Cells were collected by centrifugation at 15,000 g for 15 min at room temperature and resuspended in 333 μ L of water. Cells were added to 667 μ L boiling sodium dodecyl sulfate (7.5%, SDS w/v) and boiled for 45 min. Suspension was then washed 5X with ddH₂O using centrifugation (15 min, 15,000 g) to isolate the sacculi pellet. The sacculi preparations were resuspended in ddH₂O before imaging and further stained with Wheat Germ Agglutinin, CF®405S WGA, 0.5 μ g ml⁻¹ final concentration.

Figure legend: The FDAA fluorescence (magenta or green) is retained in isolated *A. excentricus* and *A. biprosthecum* sacculi, which are also stained with WGA lectin (cyan). Representative images are shown. Left to right: Phase, WGA, FDAA, and merge images. Scale bars: 2 μm .

- The authors suggest that differentially regulated elongasome activity (e.g. PBP2 localization) is the underlying reason for the distinct cell elongation patterns. Unfortunately, they provide only very limited experimental support and mechanistic insights to this end. To solidify this point I encourage the authors to perform additional experiments.

Previously, A22 has been used to inhibited MreB polymerization in *C. crescentus* (PMID: 22505677). Adding such a piece of data would strongly support the mecillinam findings. More generally, localization of the cytoskeletal network (MreB) or further elongasome components (RodA) would buttress the differences in PBP2 localization. Fluorescent fusion to these protein have been successfully engineered in *C. crescentus* (PMID: 22505677, PMID: 30707707).

Genetic or biochemical data are lacking altogether. Is RodA-PBP2 essential in *A. excentricus* or can PBP2 interact with alternative GT-enzymes. Generating active site point mutations in RodA or PBP2 as well as elongasome depletion strains (RodA, PBP2, MreB, MreC, MreD) and comparing their terminal phenotype could further support the authors hypothesis. Alternatively, B2H screens could be performed to demonstrate interaction of the elongasome components.

Last, additional bioinformatic analysis such as amino acid sequence comparison or AlphaFold could reveal structural differences or unique domains between the different elongasome components which might point out sites for alternative regulation or conserved residues.

We appreciate the reviewer's suggestions for further experiments to investigate the regulation of elongasome activity. However, we agree with the editor that these

additional experiments are not necessary for the scope of this study. Our current data provide sufficient support for our conclusions, and further genetic, biochemical, and bioinformatic analyses would be best addressed in future studies. As suggested below, we toned down some conclusions.

Minor comments:

- When zooming in on micrographs it seems that all microscopy images have been interpolated in the pdf file. Please check if microscopy images are of sufficient quality.

We have carefully checked all microscopy images, and they are of sufficient quality. No interpolation issues were identified in the original files.

- The red/green color scheme is not color blind friendly. Consider replacing with green/magenta or cyan/yellow.

Corrected. We have updated all figures to a green/magenta color scheme and adjusted the text accordingly.

- I would encourage the author to show more zoomed in views of single cells. While I appreciate to see multiple cells at once, it is unfortunately often hard to discern the staining pattern. Given that the authors generally show demographs and density plots with high N numbers, I think this would be totally fine and make it easier for the reader to see what the authors want to point out.

Corrected for some cells with FDAA patterns.

- Kymograph montages in Figure 1d, 6b,f; S1, S6b, S7b are all lacking a scale bar. Please include.

Done

- All demographs and heatmaps lack a display of the intensity values in their lookup table. This is particularly important for non-linear LUTs such as 'Fire' (Fig. 3a). Moreover, while the heatmap and the micrograph show a clear enrichment at mid-cell for PBP2 in *A. excentricus*, this is far less obvious in the demograph. This seems to be based on the normalization of the signal. Consider showing raw intensities.

We appreciate the reviewer's suggestion. We have now included the display of intensity values in all demographs and heatmaps to improve clarity.

Regarding the concern that PBP2 enrichment at midcell appears less obvious in the demograph compared to the heatmap and micrograph, this difference arises because the heatmap represents the localization of maxima detected via MicrobeJ, as described in the methods (lines 607–610), while the demographs display the full fluorescence signal. The heatmap approach highlights the most prominent localization sites, whereas the demographs provide a broader view of fluorescence distribution across the entire cell

body. We chose to include both analyses to provide a more comprehensive representation of PBP2 localization.

- PBP2 localization: Why do the authors use different fluorescent molecules to localize the same protein in different organisms? If possible, the authors should check whether swapping the fluorescent molecule does affect the staining pattern. Also since the author fix the cells in 70% EtOH (which denaturants proteins), I wonder if this might affect GFP and mCherry fusions to different degrees.

We appreciate the reviewer's comment. We constructed both GFP and mCherry fusions to PBP2 and observed that both behaved similarly. However, we chose to present the mCherry fusion as it exhibited stronger fluorescence.

Regarding the concern about fixation in 70% ethanol potentially affecting mCherry differently, cells were only fixed in the case of short pulse analysis with FDAA to simultaneously visualize FDAA and PBP2 localization. Control experiments for PBP2 localization were not performed under fixation conditions, as all other PBP2 localization experiments were conducted in live cells.

- I appreciate that the authors whole-genome-sequenced their mCh-PBP2 and report on the point mutations in FtsW, however this is not mentioned at any point in the main text or any reference is made to the additional text/figure.

I would also encourage the authors to add phase-contrast images of wt vs mCh-PBP2 cells along with their quantifications (Figure S8a). Furthermore, WB should be provided to demonstrate the protein is expressed as a full-length fusion.

We appreciate the reviewer's suggestions. We have now mentioned the additional figure in the main text (line 236). Additionally, we have included phase-contrast images of WT vs. mCh-PBP2 cells along with their quantifications in Extended Data Figure 8a. Finally, we have added a western blot to confirm that the mCh-PBP2 fusion is expressed as a full-length protein in the extended Data Figure 8b.

- LL251-253: As outlined above, I would encourage the authors to perform further experiments or tone this statement down.

We appreciate the reviewer's suggestion. We have toned down the statement in lines 265-267 to reflect the current level of experimental support.

- LL451: Primer sequence should be made available in a supplementary table.

Done, primer sequences are available in Table S3.

- The authors should make their quantification more homogenous, e.g. cell length and width are once represented as dot blots (Figure S8a), and violin plots (Figure S8c), while fluorescence intensity is represented as bar graphs (Figure 4d). I suggest using dot plots (not black symbols) so that mean/median above the data points can be easily seen.

We appreciate the suggestion. We have updated all quantifications to dot plots for consistency and improved readability.

RESPONSE TO REVIEWERS' COMMENTS

Reviewer #3 (Remarks to the Author):

I have read the revised version of the article by Delaby and most of my comments were either fully addressed or declined with reasonable explanation. I thus recommend this article for publication although I still have a minor stylistic comment which should be addressed prior to copy editing:

The microscopy images in the manuscript are still interpolated. I checked the figshare link and have provided a screen shot of Fig. 3a to the editor (which he can share with you) to illustrate this. While the original micrographs (center, opened in FIJI) are fine (although WGA signal is saturated!), they appear interpolated in both the summarized reviewer pdf (left), as well as the separate SVG source file (right). The same is true for the bioRxiv version of this manuscript. This can be easily seen by the absence of discernable pixels in the image. Supplementary Fig. 8 is another clear example. One easy way to circumvent this would be to scale the crops 4x4 (without interpolation) and subsequently saving as tiff files. This can easily be done in FIJI and will result in much nicer microscopy images without interpolation artefacts. Seeing the pixels at this magnification is a good sign!

While this doesn't change any of the conclusion of this paper, I just think it's a shame that a such beautiful story is hampered by less-than-ideal image rendering. This is paper is going to be well received by the community, highly cited and will certainly make its way in many lectures, potentially textbooks. I would use the chance to correct this now.

We thank Reviewer 3 for their very supportive feedback and for drawing our attention to the remaining issue with microscopy image rendering. As noted, some panels appeared interpolated due to resolution loss introduced during resizing and export in PowerPoint. Following the reviewer's helpful suggestion, we reprocessed all images using Fiji and finalized the figure layout in Adobe Illustrator. These changes preserve pixel integrity and eliminate smoothing artifacts, ensuring optimal image quality.

compiled pdf
Merge + WGA

mC
Hea

Original

SVG

Merge + WGA

183 cells

Please zoom in to see the difference!

Similar Sized image exported 4x4 scaled (no interpolation), saved as RGB tiff, and converted to CYMK pdf through Adobe Illustrator (high quality print pdf setting)

Oversized image for clarity